# Optimal and Practical Algorithms for Smooth and Strongly Convex Decentralized Optimization

**Dmitry Kovalev**     **Adil Salim**     **Peter Richtárik**

King Abdullah University of Science and Technology
Thuwal, Saudi Arabia

## Abstract

We consider the task of decentralized minimization of the sum of smooth strongly convex functions stored across the nodes of a network. For this problem, lower bounds on the number of gradient computations and the number of communication rounds required to achieve $\varepsilon$ accuracy have recently been proven. We propose two new algorithms for this decentralized optimization problem and equip them with complexity guarantees. We show that our first method is optimal both in terms of the number of communication rounds and in terms of the number of gradient computations. Unlike existing optimal algorithms, our algorithm does not rely on the expensive evaluation of dual gradients. Our second algorithm is optimal in terms of the number of communication rounds, without a logarithmic factor. Our approach relies on viewing the two proposed algorithms as accelerated variants of the Forward Backward algorithm to solve monotone inclusions associated with the decentralized optimization problem. We also verify the efficacy of our methods against state-of-the-art algorithms through numerical experiments.

## 1  Introduction

In this paper we are concerned with the design and analysis of new efficient algorithms for solving optimization problems in a *decentralized* storage and computation regime. In this regime, a network of agents/devices/workers, such as mobile devices, hospitals, wireless sensors, or smart home appliances, collaborates to solve a single optimization problem whose description is stored across the nodes of the network. Each node can perform computations using its local state and data, and is only allowed to communicate with its neighbors.

Problems of this form have been traditionally studied in the signal processing community (Xu et al., 2020), but are attracting increasing interest from the machine learning and optimization community as well (Scaman et al., 2017). Indeed, the training of supervised machine learning models via empirical risk minimization from training data stored across a network is most naturally cast as a decentralized optimization problem. Finally, while current federated learning (Konečný et al., 2016; McMahan et al., 2017) systems rely on a star network topology, with a trusted server performing aggregation and coordination placed at the center of the network, advances in decentralized optimization could be useful in new generation federated learning formulations that would rely on fully decentralized computation (Li et al., 2019). In summary, decentralized optimization is of direct relevance to machine learning, present and future.

## 1.1 Formalism

Formally, given an undirected connected network $\mathcal{G} = (\mathcal{V}, \mathcal{E})$ with nodes/vertices $\mathcal{V} = \{1, \ldots, n\}$ and edges $\mathcal{E} \subset \mathcal{V} \times \mathcal{V}$, we consider optimization problems of the form

$$\min_{x \in \mathbb{R}^d} \sum_{i \in \mathcal{V}} f_i(x), \tag{1}$$

where the data describing functions $f_i : \mathbb{R}^d \to \mathbb{R}$ is stored on node $i$ and not directly available to any other node. Decentralized algorithms for solving this problem need to respect the network structure of the problem, which is to say that computation can only be made on the nodes $i \in \mathcal{V}$ from data and information available on the nodes, and communication is constrained to only happen along the edges $e \in \mathcal{E}$.

## 1.2 Computation and communication

Several decentralized gradient-type algorithms have been proposed to solve (1) in the smooth and strongly convex regime. Two key efficiency measures used to compare such methods are: i) the *number of gradient evaluations* (where one gradient evaluation refers to computing $\nabla f_i(x_i)$ for all $i \in \mathcal{V}$ for some input vectors $x_i$), and ii) the *number of communication rounds*, where one round allows each node to send $\mathcal{O}(1)$ vectors of size $d$ to their neighbors. If computation is costly, the first comparison metric is more important, and if communication is costly, the second is more important.

Note that problem (1) poses certain intrinsic difficulties each method designed for it needs to address. Clearly, more information can be communicated in each communication round if the network $\mathcal{G}$ is "more highly" connected. By $\chi$ we denote the *condition number* associated with (the connectivity of) the graph $\mathcal{G}$; a formal definition is given later. Likewise, more computation will be needed if the functions $f_i$ are "more complicated". We will entirely focus on problems where all functions $f_i$ are $L$-smooth and $\mu$-strongly convex, which naturally leads to the quantity $\kappa := L/\mu$ as a condition number associated with computation.

Much of decentralized optimization research is focused on designing decentralized algorithms with computation and communication guarantees which have as good as possible dependence on the intrinsic properties of the problem, i.e., on the condition numbers $\kappa$ and $\chi$.

# 2 Related Work and Contributions

In this section we first briefly review some of the key results on decentralized optimization, and subsequently provide a brief summary of our key contributions.

## 2.1 Related work

Existing gradient-type decentralized methods for solving problem (1) can be informally classified into three classes: *non-accelerated* algorithms, *accelerated* algorithm and *optimal* algorithms.

**Non-accelerated methods.** Loosely speaking, a method is non-accelerated if it has at least a linear dependence on the condition numbers $\kappa$ and $\chi$, i.e., $\mathcal{O}(\kappa)$ and $\mathcal{O}(\chi)$. Please refer to (Xu et al., 2020, Table 1) for a summary of many such methods, see also (Alghunaim et al., 2019; Li and Lin, 2020). Xu et al. (2020) provide a tight unified analysis of many of these nonaccelerated algorithms, and relies on similar tools as those used in this paper, such as operator splitting and Chebyshev acceleration.

**Accelerated methods.** Accelerated methods have an improved (sublinear) dependence on the condition numbers, typically $\mathcal{O}(\sqrt{\kappa})$ and $\mathcal{O}(\sqrt{\chi})$. Accelerated algorithms include accelerated DNGD of Qu and Li (2020) and accelerated EXTRA of Li and Lin (2020); the latter using the Catalyst (Lin et al., 2017) framework to accelerate EXTRA (Shi et al., 2015). Additional accelerated methods include, the Accelerated Penalty Method of Li et al. (2018); Dvinskikh et al. (2019), SSDA and MSDA of Scaman et al. (2017) and Accelerated Dual Ascent of Uribe et al. (2020).

**Optimal algorithms.** Scaman et al. (2017) provide *lower bounds* for the gradient computation and communication complexities of finding an $\varepsilon$-accurate solution; see Section 3.2 below. There have

been several attempts to match these lower bounds, which include algorithms summarized in Table 1. Note, that gradient computation complexity is left as N/A for SSDA and MSDA. This is because they rely on the computation of the gradient of the Fenchel conjugate of $f_i$, called *dual gradients* in the sequel, which can be intractable. Indeed, computing a dual gradient can be as hard as minimizing $f_i$. Finally, we remark that Scaman et al. (2018) provide lower bounds in the nonsmooth regime as well, and an algorithm matching this lower bound is called MSPD. MSPD is primal dual Chambolle and Pock (2011), similarly to the algorithms developed in this paper.

Table 1: Comparison of existing state of the art decentralized algorithms with our results in terms of gradient computation and communication complexity of finding $x$ such that $\|x - x^*\|^2 \leq \varepsilon$, where $x^*$ is a solution to Problem (1)

| Algorithm | Gradient computation complexity | Communication complexity |
|---|---|---|
| **Existing State of the art Decentralized Algorithms** | | |
| Accelerated Dual Ascent Uribe et al. (2020) | $\mathcal{O}\left(\kappa\sqrt{\chi}\log^2\frac{1}{\varepsilon}\right)$ | $\mathcal{O}\left(\sqrt{\kappa\chi}\log\frac{1}{\varepsilon}\right)$ |
| Single/Multi Step Dual Ascent Scaman et al. (2017) | N/A | $\mathcal{O}\left(\sqrt{\kappa\chi}\log\frac{1}{\varepsilon}\right)$ |
| Accelerated Penalty Method Li et al. (2018); Dvinskikh et al. (2019) | $\mathcal{O}\left(\sqrt{\kappa}\log\frac{1}{\varepsilon}\right)$ | $\mathcal{O}\left(\sqrt{\kappa\chi}\log^2\frac{1}{\varepsilon}\right)$ |
| Accelerated EXTRA Li and Lin (2020) | $\mathcal{O}\left(\sqrt{\kappa\chi}\log(\kappa\chi)\log\frac{1}{\varepsilon}\right)$ | $\mathcal{O}\left(\sqrt{\kappa\chi}\log(\kappa\chi)\log\frac{1}{\varepsilon}\right)$ |
| **Our Results** | | |
| Algorithm 1 this paper, Theorem 2 | $\mathcal{O}\left(\left(\sqrt{\kappa\chi}+\chi\right)\log\frac{1}{\varepsilon}\right)$ | $\mathcal{O}\left(\left(\sqrt{\kappa\chi}+\chi\right)\log\frac{1}{\varepsilon}\right)$ |
| **Algorithm 2 this paper, Corollary 1** | $\mathcal{O}\left(\sqrt{\kappa}\log\frac{1}{\varepsilon}\right)$ | $\mathcal{O}\left(\sqrt{\kappa\chi}\log\frac{1}{\varepsilon}\right)$ |
| Algorithm 3 this paper, Appendix | $\mathcal{O}\left(\sqrt{\kappa\chi}\log\frac{1}{\varepsilon}\right)$ | $\mathcal{O}\left(\sqrt{\kappa\chi}\log\frac{1}{\varepsilon}\right)$ |
| **Lower bounds Scaman et al. (2017)** | $\mathcal{O}\left(\sqrt{\kappa}\log\frac{1}{\varepsilon}\right)$ | $\mathcal{O}\left(\sqrt{\kappa\chi}\log\frac{1}{\varepsilon}\right)$ |

## 2.2 Summary of contributions

The starting point of this paper is the realization that, *to the best of our knowledge, in the class of algorithms not relying on the computation of the dual gradients, there is no algorithm optimal in communication complexity, and as a result, no algorithm optimal in both gradient computation and communication complexity.* To remedy this situation, we do the following:

- We propose a new accelerated decentralized algorithm not relying on dual gradients: Accelerated Proximal Alternating Predictor-Corrector (APAPC) method (Algorithm 1). We show that in order to obtain $x$ for which $\|x - x^*\|^2 \leq \varepsilon$, where $x^*$ is the solution of (1), this method only needs

$$\mathcal{O}((\sqrt{\kappa\chi}+\chi)\log(1/\varepsilon))$$

  gradient computations and communication rounds (Theorem 2). When combined with Chebyshev acceleration, similarly to the trick used in (Scaman et al., 2017, Section 4.2), we show that our method, which we then call Optimal Proximal Alternating Predictor-Corrector (OPAPC) method (Algorithm 2), leads to an *optimal* decentralized method both in terms of gradient computation and communication complexity (Corollary 1). In particular, OPAPC finds an $\varepsilon$-solution in at most

$$\mathcal{O}\left(\sqrt{\kappa}\log(1/\varepsilon)\right)$$

  gradient computations and at most

$$\mathcal{O}\left(\sqrt{\kappa\chi}\log(1/\varepsilon)\right)$$

communication rounds. Algorithm 2 reaches the lower bounds (Theorem 1), and hence it is indeed optimal.

- We also propose another accelerated algorithm (Algorithm 3) not relying on dual gradients, one that is optimal in communication complexity (this algorithm is presented in the appendix only). Compared to the above development, this algorithm has the added advantage that it requires the computation of a single gradient per communication step. This can have practical benefits when communication is expensive.

## 3 Background

### 3.1 Basic formulation of the decentralized problem

Problem (1) can be reformulated as a lifted (from $\mathbb{R}^d$ to $\mathbb{R}^{dn}$) optimization problem via consensus constraints:

$$\min_{\substack{x_1,\dots,x_n \in \mathbb{R}^d \\ x_1=\dots=x_n}} \sum_{i \in \mathcal{V}} f_i(x_i). \tag{2}$$

Consider the function $F : (\mathbb{R}^d)^{\mathcal{V}} \to \mathbb{R}$ defined by $F(x_1,\dots,x_n) = \sum_{i \in \mathcal{V}} f_i(x_i)$, where $x_1,\dots,x_n \in \mathbb{R}^d$. Then, $F$ is $\mu$-strongly convex and $L$-smooth since the individual functions $f_i$ are. Consider also any linear operator (equivalently, any matrix) $\mathbf{W} : (\mathbb{R}^d)^{\mathcal{V}} \to (\mathbb{R}^d)^{\mathcal{V}}$ such that $\mathbf{W}(x_1,\dots,x_n) = 0$ if and only if $x_1 = \dots = x_n$. Denoting $x = (x_1,\dots,x_n) \in (\mathbb{R}^d)^{\mathcal{V}}$, Problem (2) is equivalent to

$$\min_{x \in \ker(\mathbf{W})} F(x). \tag{3}$$

Many optimization algorithms converge exponentially fast (i.e., linearly) to a minimizer of Problem (3), e.g. the projected gradient algorithm. However, only few of them are decentralized. A decentralized algorithm typically relies on multiplication by $\mathbf{W}$, in cases where $\mathbf{W}$ is a *gossip matrix*. Consider a $n \times n$ matrix $\hat{\mathbf{W}}$ satisfying the following properties: 1) $\hat{\mathbf{W}}$ is symmetric and positive semi definite, 2) $\hat{\mathbf{W}}_{i,j} \neq 0$ if and only if $i = j$ or $(i,j) \in \mathcal{E}$, and 3) $\ker \hat{\mathbf{W}} = \mathrm{span}(\mathbb{1})$, where $\mathbb{1} = (1,\dots,1)^{\top}$. Such a matrix is called a *gossip matrix*. A typical example is the Laplacian of the graph $\mathcal{G}$. Denoting $\mathbf{I}$ the $d \times d$ identity matrix and $\otimes$ the Kronecker product, consider $\mathbf{W} : (\mathbb{R}^d)^{\mathcal{V}} \to (\mathbb{R}^d)^{\mathcal{V}}$ the $nd \times nd$ matrix defined by $\mathbf{W} := \hat{\mathbf{W}} \otimes \mathbf{I}$. This matrix can be represented as a block matrix $\mathbf{W} = (\mathbf{W}_{i,j})_{(i,j) \in \mathcal{V}^2}$, where each block $\mathbf{W}_{i,j} = \hat{\mathbf{W}}_{i,j}\mathbf{I}$ is a $d \times d$ matrix proportional to $\mathbf{I}$. In particular, if $d = 1$, then $\mathbf{W} = \hat{\mathbf{W}}$. Moreover, $\mathbf{W}$ satisfy similar properties to $\hat{\mathbf{W}}$:

1. $\mathbf{W}$ is symmetric and positive semi definite,
2. $\mathbf{W}_{i,j} \neq 0$ if and only if $i = j$ or $(i,j) \in \mathcal{E}$,
3. $\ker \mathbf{W}$ is the consensus space, $\ker(\mathbf{W}) = \{(x_1,\dots,x_n) \in (\mathbb{R}^d)^{\mathcal{V}}, x_1 = \dots = x_n\}$,
4. $\lambda_{\max}(\mathbf{W}) = \lambda_{\max}(\hat{\mathbf{W}})$ and $\lambda_{\min}^{+}(\mathbf{W}) = \lambda_{\min}^{+}(\hat{\mathbf{W}})$, where $\lambda_{\max}$ (resp. $\lambda_{\min}^{+}$) denotes the largest (resp. the smallest positive) eigenvalue.

Throughout the paper, we denote $\mathbf{W}^{\dagger} : \mathrm{range}(\mathbf{W}) \to \mathrm{range}(\mathbf{W})$ the inverse of the map $\mathbf{W} : \mathrm{range}(\mathbf{W}) \to \mathrm{range}(\mathbf{W})$. The operator $\mathbf{W}^{\dagger}$ is positive definite over $\mathrm{range}(\mathbf{W})$ and we denote $\|y\|_{\mathbf{W}^{\dagger}}^2 = \langle \mathbf{W}^{\dagger}y, y \rangle$ for every $y \in \mathrm{range}(\mathbf{W})$. With a slight abuse of language, we shall say that $\mathbf{W}$ is a gossip matrix. Note that decentralized communication can be represented as a multiplication of $\mathbf{W}$ by a vector $x \in (\mathbb{R}^d)^{\mathcal{V}}$. Indeed, the $i^{\text{th}}$ component of $\mathbf{W}x$ is a linear combination of $x_j$, where $j$ is a neighbor of $i$ (we shall write $j \sim i$). In other words, one matrix vector multiplication involving $\mathbf{W}$ is equivalent to one communication round.

*In the rest of the paper, our goal is to solve the equivalent problem* (3) *with* $\mathbf{W}$ *being a gossip matrix via an optimization algorithm which uses only evaluations of* $\nabla F$ *and multiplications by* $\mathbf{W}$.

### 3.2 Lower bounds

Linearly converging decentralized algorithms using a gossip matrix $\mathbf{W}$ often have a linear rate depending on the condition number of the $f_i$, $\kappa := \frac{L}{\mu}$ and the condition number (or spectral gap) of

$\mathbf{W}$, $\chi(\mathbf{W}) \coloneqq \frac{\lambda_{\max}(\mathbf{W})}{\lambda_{\min}^+(\mathbf{W})}$. Indeed, the spectral gap of the Laplacian matrix is known to be a measure of the connectivity of the graph.

In this paper, we define *the class of (first order) decentralized algorithms* as the subset of black box optimization procedure (Scaman et al., 2017, Section 3.1) not using dual gradients, i.e. a decentralized algorithm is not allowed to compute $\nabla f_i^*$ (a formal definition is given in the Supplementary material). Complexity lower bounds for solving Problem (1) by a black-box optimization procedure are given by Scaman et al. (2017). These lower bounds relate the number of gradient computations (resp. number of communication rounds) to achieve $\varepsilon$ accuracy to the condition numbers $\kappa$ and $\chi(\mathbf{W})$. Since a decentralized algorithm is a black-box optimization procedure, these lower bounds apply to decentralized algorithms. Therefore, we obtain our first result as a direct application of (Scaman et al., 2017, Corollary 2).

**Theorem 1** (Scaman et al. (2017)). *Let $\chi \geq 1$. There exist a gossip matrix $\mathbf{W}$ with condition number $\chi$, and a family of smooth strongly convex functions $(f_i)_{i \in \mathcal{V}}$ with condition number $\kappa > 0$ such that the following holds: for any $\varepsilon > 0$, any decentralized algorithm requires at least $\Omega\left(\sqrt{\kappa\chi}\log(1/\varepsilon)\right)$ communication rounds, and at least $\Omega\left(\sqrt{\kappa}\log(1/\varepsilon)\right)$ gradient computations to output $x = (x_1, \ldots, x_n)$ such that $\|x - x^*\|^2 < \varepsilon$, where $x^* = \arg\min F$.*

Although the lower bounds of Theorem 1 are obvious consequences of (Scaman et al., 2017, Corollary 2), their tightness is not. Indeed, the lower bounds of Theorem 1 are tight on the class of black-box optimization procedures since they are achieved by MSDA Scaman et al. (2017). However, MSDA uses dual gradients and whether these lower bounds are tight on the class of decentralized algorithms is not known. In this paper, we propose decentralized algorithms achieving these lower bounds, showing in particular that they are tight.

### 3.3 Operator splitting

Recall that in this paper, any optimization algorithm solving Problem (3) by using evaluations of $\nabla F$ and multiplications by the gossip matrix $\mathbf{W}$ only is a decentralized algorithm. Such algorithms can be obtained in several ways, e.g., by applying operator splitting methods to primal dual reformulations of Problem (3), see Condat et al. (2019). This is the approach we chose in this work.

We now provide some minimal background knowledge on the Forward Backward algorithm involving monotone operators. We restrict ourselves to single valued, continuous monotone operators. For the general case of set valued monotone operators, the reader is referred to Bauschke and Combettes (2011).

Let $\mathsf{E}$ be an Euclidean space and denote $\langle \cdot, \cdot \rangle_\mathsf{E}$, $\|\cdot\|_\mathsf{E}$ its inner product and the associated norm. Given $\nu \in \mathbb{R}$, a map $A : \mathsf{E} \to \mathsf{E}$ is *$\nu$-monotone* if for every $x, y \in \mathsf{E}$,

$$\langle A(x) - A(y), x - y \rangle_\mathsf{E} \geq \nu \|x - y\|_\mathsf{E}^2.$$

If $\nu < 0$, $A$ is *weakly monotone*, if $\nu > 0$, $A$ is *strongly monotone* and if $\nu = 0$ then $A$ is *monotone*. In this paper, a monotone operator is defined as a monotone continuous map. For every monotone operator and every $\gamma > 0$, the map $I + \gamma A : \mathsf{E} \to \mathsf{E}$ is one-to-one and its inverse $J_{\gamma A} = (I + \gamma A)^{-1} : \mathsf{E} \to \mathsf{E}$, called *resolvent*, is well defined. Let $F$ be a *smooth* convex function, i.e., $F$ is differentiable and its gradient is Lipschitz continuous. Then $\nabla F$ is a monotone operator, and the resolvent $J_{\gamma \nabla F}$ is the proximity operator of $\gamma F$. However, there exist monotone operators which are not gradients of convex functions. For instance, a skew symmetric operator $S$ on $\mathsf{E}$ defines the linear map $x \mapsto Sx$ which is not a gradient. This map is a monotone operator since $\langle Sx, x \rangle_\mathsf{E} = 0$. The *set of zeros* of $A$, defined as $Z(A) \coloneqq \{x \in \mathsf{E}, A(x) = 0\}$, is often of interest in optimization. For instance, $Z(\nabla F) = \arg\min F$.

**Forward Backward.** In order to find an element in $Z(A + B)$, where $B$ is another monotone operator, the *Forward Backward algorithm* iterates

$$x^{k+1} = J_B(x^k - A(x^k)). \tag{4}$$

Note that if $A = \nabla F$ and $B = \nabla G$, where $G$ is another differentiable convex function, the Forward Backward algorithm boils down to the proximal gradient algorithm. In this particular case, Nesterov acceleration can be applied to (4) and leads to faster convergence rates compared to the proximal gradient algorithm (Nesterov, 1983; Beck and Teboulle, 2009).

**Generalized Forward Backward.**   For every positive definite operator $\mathbf{P}$ on E, the algorithm

$$x^{k+1} = J_{\mathbf{P}^{-1}B}(x^k - \mathbf{P}^{-1}A(x^k)), \qquad (5)$$

called the *Generalized Forward Backward* method, can be seen as an instance of (4) because $Z(\mathbf{P}^{-1}A + \mathbf{P}^{-1}B) = Z(A+B)$ and $\mathbf{P}^{-1}A$, $\mathbf{P}^{-1}B$ are monotone operators under the inner product induced by $\mathbf{P}$ on E. For example, the gradient of $F$ under this inner product is $\mathbf{P}^{-1}\nabla F$. A primal dual optimization algorithm is an algorithm solving a primal dual formulation of a minimization problem, see below. Many primal dual algorithms can be seen as instances of (5), with general monotone operators $A$, $B$, for a well chosen parameter $\mathbf{P}$, see (Condat et al., 2019).

## 4   New Decentralized Algorithms

### 4.1   An accelerated primal dual algorithm

Before presenting our algorithm, we introduce an accelerated decentralized algorithm which we then use to motivate the development of our method.

In this section, E is the Euclidean space $\mathsf{E} = (\mathbb{R}^d)^{\mathcal{V}} \times \text{range}(\mathbf{W})$ endowed with the norm $\|(x,y)\|_{\mathsf{E}}^2 := \|x\|^2 + \|y\|_{\mathbf{W}^{\dagger}}^2$.

Using the first order optimality conditions, a point $x^*$ is a solution to Problem (3) if and only if $\nabla F(x^*) \in \text{range}(\mathbf{W})$ and $x^* \in \text{ker}(\mathbf{W})$. Solving Problem (3) is thus equivalent to finding $(x^*, y^*) \in \mathsf{E}$ such that

$$\begin{aligned} 0 &= \nabla F(x^*) + y^*, \\ 0 &= \mathbf{W}x^*. \end{aligned} \qquad (6)$$

Indeed, the first line of (6) is equivalent to $\nabla F(x^*) = -y^* \in \text{range}\mathbf{W}$, because $(x^*, y^*) \in \mathsf{E} = (\mathbb{R}^d)^{\mathcal{V}} \times \text{range}(\mathbf{W})$. The second line of (6) is just a definition of $x^* \in \text{ker }\mathbf{W}$. Consider the maps $M, A, B : \mathsf{E} \to \mathsf{E}$ defined by

$$M(x,y) := \begin{bmatrix} \nabla F(x) + y \\ -\mathbf{W}x \end{bmatrix}, \quad A(x,y) := \begin{bmatrix} \nabla F(x) \\ 0 \end{bmatrix}, \quad B(x,y) := \begin{bmatrix} y \\ -\mathbf{W}x \end{bmatrix}.$$

Then $M, A$ and $B$ are monotone operators. Indeed, $A$ is the gradient of the convex function $(x,y) \mapsto F(x)$, $B$ satisfies

$$\langle B(x,y),(x,y)\rangle_{\mathsf{E}} = \langle x - \mathbf{W}^{\dagger}\mathbf{W}x, y\rangle = 0$$

for every $(x,y) \in \mathsf{E}$ (since $y \in \text{range}(\mathbf{W})$), and $M = A + B$. Moreover, $M(x^*, y^*) = 0$, i.e., $(x^*, y^*)$ is a zero of $M$.

One idea to solve (6) is therefore to apply Algorithm (4) to the sum $A + B$. However, computing the resolvent $J_B$ in a decentralized way across the network $\mathcal{G}$ is notably challenging. Another idea is to apply (5) using the symmetric positive definite operator $P : \mathsf{E} \to \mathsf{E}$ defined by

$$\mathbf{P} = \begin{bmatrix} \frac{1}{\eta}\mathbf{I} & 0 \\ 0 & \frac{1}{\theta}\mathbf{I} - \eta\mathbf{W} \end{bmatrix}.$$

Indeed, for every $(x,y) \in \mathsf{E}$, $(x',y') = J_{P^{-1}B}(x,y)$ implies $x' = x - \eta y'$ and $\frac{1}{\theta}(y'-y) - \eta\mathbf{W}(y' - y) = \mathbf{W}x' = \mathbf{W}(x - \eta y')$. Therefore, $y' = y + \theta\mathbf{W}(x - \eta y)$, and the computation of $J_{P^{-1}B}$ only requires one multiplication by $\mathbf{W}$, i.e., one local communication round. The resulting algorithm is

$$\begin{aligned} y^{k+1} &= y^k + \theta\mathbf{W}(x^k - \eta\nabla F(x^k) - \eta y^k), \\ x^{k+1} &= x^k - \eta\nabla F(x^k) - \eta y^{k+1}. \end{aligned} \qquad (7)$$

**Remark 1.** *The Proximal Alternating Predictor–Corrector (PAPC) algorithm, a.k.a. Loris–Verhoven (Loris and Verhoeven, 2011; Drori et al., 2015; Chen et al., 2013; Condat et al., 2019) is a primal dual algorithm that can tackle Problem* (3). *Up to a change of variable, Algorithm* (7) *can be shown to be equivalent to PAPC applied to* (3). *Moreover, it was already noticed that the PAPC can be represented as a Forward Backward algorithm* (5) *(Condat et al., 2019).*

Invoking a complexity result on the PAPC from Salim et al. (2020), the complexity of Algorithm (7) is $\mathcal{O}\left((\kappa + \chi(\mathbf{W})) \log(1/\varepsilon)\right)$, both in communication and gradient computations. This complexity is equivalent to that of the best performing non accelerated algorithm proposed recently, such as Exact diffusion, NIDS and EXTRA (see Li and Lin (2020); Xu et al. (2020)). In spite of this, we are able to accelerate the convergence of Algorithm (7).

In particular, we propose a new algorithm that can be seen as an accelerated version of Algorithm (7). The proposed algorithm (APAPC) is defined in Algorithm 1), and its complexity is given in Theorem 2. We prove that the complexity of APAPC is $\mathcal{O}((\sqrt{\kappa\chi(\mathbf{W})}+\chi(\mathbf{W})) \log(1/\varepsilon))$, both in communication rounds and gradient computations. The proposed algorithm is accelerated because its dependence on the condition number $\kappa$ is $\mathcal{O}(\sqrt{\kappa})$ instead of $\mathcal{O}(\kappa)$.

---

**Algorithm 1** Accelerated PAPC (APAPC)

---

1: **Parameters:** $x^0 \in \mathbb{R}^{nd}, y^0 \in \mathrm{range}\mathbf{W}, \eta, \theta, \alpha > 0, \tau \in (0, 1)$
2: Set $x_f^0 = x^0$
3: **for** $k = 0, 1, 2, \ldots$ **do**
4: $\quad x_g^k = \tau x^k + (1 - \tau)x_f^k$
5: $\quad x^{k+1/2} = (1 + \eta\alpha)^{-1}(x^k - \eta(\nabla F(x_g^k) - \alpha x_g^k + y^k))$
6: $\quad y^{k+1} = y^k + \theta\mathbf{W}x^{k+1/2}$
7: $\quad x^{k+1} = (1 + \eta\alpha)^{-1}(x^k - \eta(\nabla F(x_g^k) - \alpha x_g^k + y^{k+1}))$
8: $\quad x_f^{k+1} = x_g^k + \frac{2\tau}{2-\tau}(x^{k+1} - x^k)$
9: **end for**

---

**Theorem 2** (Accelerated PAPC). *Set the parameters $\eta, \theta, \alpha, \tau$ to $\eta = \frac{1}{4\tau L}$, $\theta = \frac{1}{\eta\lambda_{\max}(\mathbf{W})}$, $\alpha = \mu$, and $\tau = \min\left\{1, \frac{1}{2}\sqrt{\frac{\chi(\mathbf{W})}{\kappa}}\right\}$. Then,*

$$\frac{1}{\eta}\left\|x^k - x^*\right\|^2 + \frac{2(1-\tau)}{\tau}\mathrm{D}_F(x_f^k, x^*) \leq \left(1 + \frac{1}{4}\min\left\{\frac{1}{\sqrt{\kappa\chi(\mathbf{W})}}, \frac{1}{\chi(\mathbf{W})}\right\}\right)^{-k} C,$$

*where $\mathrm{D}_F$ is the Bregman divergence of $F$ and $C := \frac{1}{\eta}\left\|x^0 - x^*\right\|^2 + \frac{1}{\theta}\|y^0 - y^*\|_{\mathbf{W}\dagger}^2 + \frac{2(1-\tau)}{\tau}\mathrm{D}_F(x_f^0, x^*)$. Moreover, for every $\varepsilon > 0$, APAPC finds $x^k$ for which $\|x^k - x^*\|^2 \leq \varepsilon$ in at most $\mathcal{O}\left(\left(\sqrt{\kappa\chi(\mathbf{W})} + \chi(\mathbf{W})\right)\log(1/\varepsilon)\right)$ computations (resp. communication rounds).*

The proposed algorithm 1 provably accelerates Algorithm (7). The proof intuitively relies on viewing Algorithm 1 as an accelerated version of (5), although Nesterov's acceleration does not apply to general monotone operators *a priori*.

## 4.2 A decentralized algorithm optimal both in communication and computation complexity

As mentioned before, while APAPC is accelerated, it is not optimal. We now derive a variant which is optimal both in gradient computations and communication rounds. Following Scaman et al. (2017, Section 4.2), our main tool to derive the new decentralized optimal algorithm is the Chebyshev acceleration (Scaman et al., 2017; Arioli and Scott, 2014).

In particular, there exists a polynomial $P$ such that

    (i) $P(\mathbf{W})$ is a Gossip matrix,

(ii) multiplication by $P(\mathbf{W})$ requires $\left\lfloor\sqrt{\chi(\mathbf{W})}\right\rfloor$ multiplications by $\mathbf{W}$ (i.e., communication rounds) and is described by the subroutine ACCELERATEDGOSSIP proposed in (Scaman et al., 2017, Algorithm 2) and recalled in Algorithm 2 for the ease of reading,

(iii) $\chi(P(\mathbf{W})) \leq 4$.

Therefore, one can replace $\mathbf{W}$ by $P(\mathbf{W})$ in Problem (3) to obtain an equivalent problem. Applying APAPC to the equivalent problem leads to a linearly converging decentralized algorithm. This new algorithm, called Optimal PAPC (OPAPC), is formalized as Algorithm 2.

---

**Algorithm 2** Optimal PAPC (OPAPC)

---

1: **Parameters:** $x^0 \in \mathbb{R}^{nd}, y^0 \in \text{range}P(\mathbf{W}), T \in \mathbb{N}^*, c_1, c_2, c_3, \eta, \theta, \alpha > 0, \tau \in (0,1)$
2: Set $x_f^0 = x^0$
3: **for** $k = 0, 1, 2, \ldots$ **do**
4:     $x_g^k = \tau x^k + (1-\tau)x_f^k$
5:     $x^{k+1/2} = (1+\eta\alpha)^{-1}(x^k - \eta(\nabla F(x_g^k) - \alpha x_g^k + y^k))$
6:     $y^{k+1} = y^k + \theta\text{ACCELERATEDGOSSIP}(\mathbf{W}, x^{k+1/2}, T)$
7:     $x^{k+1} = (1+\eta\alpha)^{-1}(x^k - \eta(\nabla F(x_g^k) - \alpha x_g^k + y^{k+1}))$
8:     $x_f^{k+1} = x_g^k + \frac{2\tau}{2-\tau}(x^{k+1} - x^k)$
9: **end for**
10: **procedure** ACCELERATEDGOSSIP$(\mathbf{W}, x, T)$
11:     Set $a_0 = 1, a_1 = c_2, x_0 = x, x_1 = c_2(I - c_3\mathbf{W})x$
12:     **for** $i = 1, \ldots, T-1$ **do**
13:         $a_{i+1} = 2c_2a_i - a_{i-1}$
14:         $x_{i+1} = 2c_2(I - c_3\mathbf{W})x_i - x_{i-1}$
15:     **end for**
        **return** $x - \frac{x_T}{a_T}$
16: **end procedure**

---

Using the properties of $P(\mathbf{W})$ mentioned above, we obtain the following corollary of Theorem 2.

**Corollary 1** (Optimal PAPC). *Set the parameters* $T, c_1, c_2, c_3, \eta, \theta, \alpha, \tau$ *to*

$$T = \left\lfloor\sqrt{\chi(\mathbf{W})}\right\rfloor, \quad c_1 = \frac{\sqrt{\chi(\mathbf{W})}-1}{\sqrt{\chi(\mathbf{W})}+1}, \quad c_2 = \frac{\chi(\mathbf{W})+1}{\chi(\mathbf{W})-1}, \quad c_3 = \frac{2\chi(\mathbf{W})}{(1+\chi(\mathbf{W}))\lambda_{\max}(\mathbf{W})},$$

$$\eta = \frac{1}{4\tau L}, \quad \theta = \frac{1+c_1^{2T}}{\eta(1+c_1^T)^2}, \quad \alpha = \mu, \quad \tau = \min\left\{1, \frac{1+c_1^T}{2\sqrt{\kappa}(1-c_1^T)}\right\}.$$

*Then, there exists* $C \geq 0$ *such that*

$$\frac{1}{\eta}\left\|x^k - x^*\right\|^2 + \frac{2-\tau}{\tau}D_F(x_f^k, x^*) \leq \left(1 + \frac{1}{16}\min\left\{\frac{2}{\sqrt{\kappa}}, 1\right\}\right)^{-k} C.$$

*Moreover, for every* $\varepsilon > 0$*, OPAPC finds* $x^k$ *for which* $\|x^k - x^*\|^2 \leq \varepsilon$ *in at most* $\mathcal{O}\left(\sqrt{\kappa}\log(1/\varepsilon)\right)$ *gradient computations and at most* $\mathcal{O}\left(\sqrt{\kappa\chi(\mathbf{W})}\log(1/\varepsilon)\right)$ *communication rounds.*

The Algorithm 2 achieves both the lower bounds of Theorem 1. In particular, the lower bounds of Theorem 1 are tight.

## 5 Numerical Experiments

In this section, we perform experiments with logistic regression for binary classification with $\ell^2$ regularizer, where our loss function has the form

$$f_i(x) = \frac{1}{m}\sum_{j=1}^{m}\log(1 + \exp(-b_{ij}a_{ij}^\top x)) + \frac{r}{2}\|x\|^2,$$

where $a_{ij} \in \mathbb{R}^d$, $b_{ij} \in \{-1, +1\}$ are data points, $r$ is the regularization parameter, $m$ is the number of data points stored on each node.

In our experiments we used $10,000$ data samples randomly distributed to the nodes of network of size $n = 100$, $m = 100$ samples per each node. We used 2 networks: $10 \times 10$ grid and Erdös-Rényi random graph of average degree 6. Same setup was tested by Scaman et al. (2017).

We use three LIBSVM[1] datasets: a6a, w6a, ijcnn1. The regularization parameter was chosen so that $\kappa \approx 10^3$. Additional experiments with synthetic data are given in the Supplementary material.

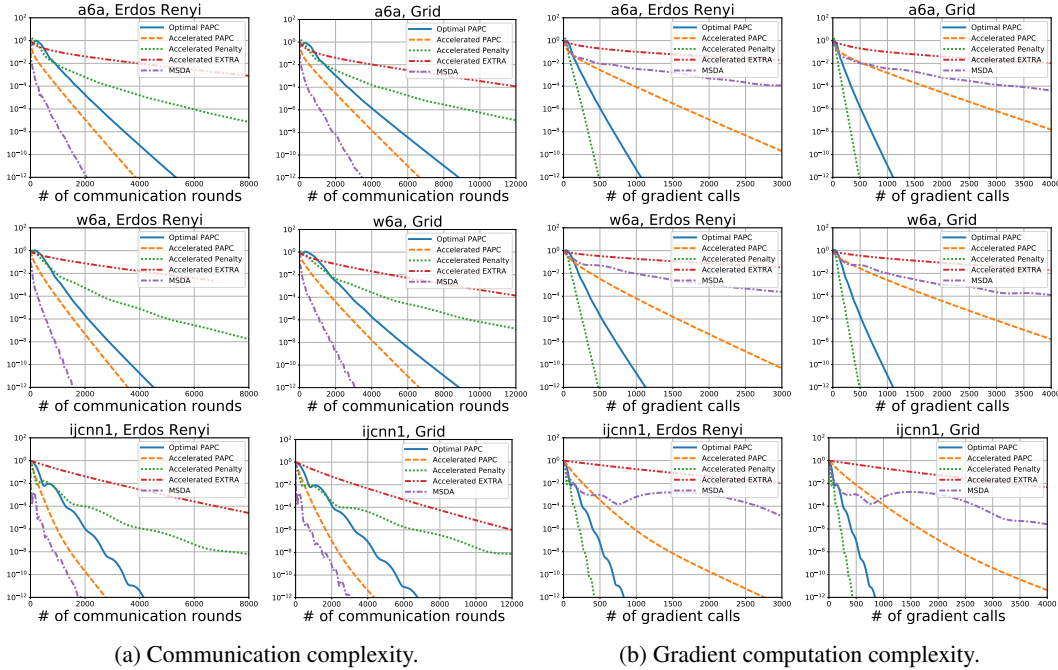

(a) Communication complexity.　　　　　　(b) Gradient computation complexity.

Figure 1: Linear convergence of decentralized algorithms in number of communication rounds and gradient computations.

Figure 1 compares Algorithm 1 (Accelerated PAPC) and Algorithm 2 (Optimal PAPC) with three state-of-the-art accelerated benchmarks: Accelerated Penalty (Li et al., 2018; Dvinskikh et al., 2019), Accelerated Extra (Li and Lin, 2020) and MSDA Scaman et al. (2017), where we used the subroutine of Uribe et al. (2020) to compute the dual gradients. This subroutine uses primal gradients $\nabla f_i$, and the resulting algorithm can be shown to have an optimal communication complexity. We represent the squared distance to the solution as a function of the number of communication rounds and (primal) gradient computations.

The theory developed in this paper concerns the value of the linear rates of the proposed algorithms, i.e., the slope of the curves in Figure 1. In communication complexity, one can see that our Algorithms 1 and 2 have similar rate and perform better than the other benchmarks except MSDA. MSDA performs slightly better in communication complexity. However, MSDA uses dual gradients and has much higher iteration complexity. In gradient computation complexity, one can see that our main Algorithm 2 is, alongside Accelerated Penalty, the best performing method. Accelerated Penalty performs slightly better in gradient computation complexity. However, the theory of Accelerated Penalty does not predict linear convergence in the number of communication rounds and we see that this algorithm converges sublinearly. Overall, Optimal PAPC is the only universal method which performs well both in communication rounds and gradient computations.

# 6 Broader Impact

Our paper is of a fundamental theoretical nature. We designed new decentralized optimization algorithms, and proved that they are optimal in a certain rigorous mathematical sense. We trust that our methods will have impact wherever decentralized optimization is needed and used, at present (e.g., sensor networks) and in the future (e.g., fully decentralized federated learning). Having said that, our methods are generic as we did not investigate any particular application. Hence, we do not expect any immediate societal impact beyond impact on the research community developing the foundational tools for AI. We hope, however, that AI practitioners will be inspired by this work and will use the fruits of our labor to benefit humanity through concrete applications and machine learning models.

## Footnotes

[1]The LIBSVM dataset collection is available at `https://www.csie.ntu.edu.tw/~cjlin/libsvmtools/datasets/`

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
