[Supplementary Material]

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

# Appendix

## Contents

# A  Experiments with synthetic data

In this section, we present additional experiments. The experimental setup is the same as before, with only one difference: we use randomly generated dataset with the following choice of the number of features $d$: 40, 60, 80, 100. The results, which are shown in Figure 2, are similar to the previous results, and the same conclusions can be made.

(a) Communication complexity.

(b) Gradient computation complexity.

Figure 2: Linear convergence of decentralized algorithms in number of communication rounds and gradient computations.

# B Formal Definition of Decentralized Algorithms

In this paper, we considered the resolution of (1) distributively across the nodes of the network $G$. Each node $i \in \mathcal{V}$ is associated with a computing agent that only have access to the local function $f_i$. The goal of the network of computing agent is to minimize the function (1) by performing local computations involving $f_i$ at each node $i$ and by communicating vectors along the edges, i.e., with neighbors $j \sim i$.

More precisely, we considered the class of decentralized algorithms, similarly to (Scaman et al., 2017, Section 3.1). In this paper, a decentralized algorithm is formally defined as an algorithm satisfying the following constraints. At time $k$, each node $i$ possesses a local internal memory $M_i^k \subset \mathbb{R}^d$ and outputs an estimation $x_i^k \in M_i^k$ of the solution to Problem (1). This internal memory is updated via gradient computations and communication rounds i.e.,

$$M_i^{k+1} \subset \mathrm{Span}(\mathrm{Comm}_i^{k+1} \bigcup \mathrm{Comp}_i^{k+1}),$$

where $\mathrm{Comm}_i^{k+1}$ is the communication component and $\mathrm{Comp}_i^{k+1}$ the computation component. The communication component is updated by combining the elements of the local memories of nodes $j \sim i$ at time $k$: $\mathrm{Comm}_i^{k+1} = \mathrm{Span}(\bigcup_{j \sim i} M_j^k)$. The computation component is updated by combining the elements of the local memory of $i$ at time $k$ along with the gradients of the local functions $f_i$ at these elements: $\mathrm{Comp}_i^{k+1} = \mathrm{Span}(\{x, \nabla f_i(x), x \in M_i^k\})$. Compared to the class of black-box optimization procedures of (Scaman et al., 2017), the class of decentralized algorithm is smaller (i.e., included). Indeed, black-box optimization procedures use dual gradients. In other words, they use the following definition of the computation component:

$$\widetilde{\mathrm{Comp}}_i^{k+1} = \{x, \nabla f_i(x), \nabla f_i^*(x), x \in M_i^k\}$$

(where $f_i^*$ is the Fenchel transform of $f_i$), which is a set containing $\mathrm{Comp}_i^{k+1}$. Recall that computing the dual gradient $\nabla f_i^*(0)$ is equivalent to minimizing $f_i$.

Finally, as in Scaman et al. (2017), we say that a decentralized algorithm uses the gossip matrix $\mathbf{W}$ if the local communication is achieved by multiplication of a vector by $\mathbf{W}$.

## C Proof of Theorem 2 (APAPC)

For every $p \geq 0$, we denote by $\|\cdot\|_\mathbf{P}$ the (semi)-norm induced by any positive (semi)-definite matrix $\mathbf{P} : \mathbb{R}^p \to \mathbb{R}^p$.

**Lemma 1.** *Let $\mathbf{P} \in \mathbb{R}^{2nd \times 2nd}$ be the following matrix:*

$$\mathbf{P} = \begin{bmatrix} \frac{1}{\eta}\mathbf{I} & 0 \\ 0 & \frac{1}{\theta}\mathbf{W}^\dagger - (1+\eta\alpha)^{-1}\eta\mathbf{I} \end{bmatrix}. \tag{8}$$

*If parameters $\eta$ and $\theta$ satisfy*

$$\eta\theta\lambda_{\max}(\mathbf{W}) \leq 1, \tag{9}$$

*then for all $x \in \mathbb{R}^{nd}$, $y \in \mathrm{range}\mathbf{W}$ the following inequality holds:*

$$\frac{1}{\eta}\|x\|^2 \leq \left\| \begin{bmatrix} x \\ y \end{bmatrix} \right\|_\mathbf{P}^2 \leq \frac{1}{\eta}\|x\|^2 + \frac{1}{\theta}\|y\|_{\mathbf{W}^\dagger}^2. \tag{10}$$

*Proof.* Note that under our assumptions, the matrix $\frac{1}{\theta}\mathbf{W}^\dagger - (1+\eta\alpha)^{-1}\eta\mathbf{I}$ is positive semi-definite on $\mathrm{range}\mathbf{W}$. $\square$

**Lemma 2.** *Let $\alpha$ satisfy $0 \leq \alpha \leq \mu$. Then the following inequality holds:*

$$-\frac{1}{2\eta}\|x^{k+1} - x^k\|^2 \leq -\frac{\eta}{4}\|y^{k+1} - y^*\|^2 + \eta\alpha^2\|x^{k+1} - x^*\|^2 + 2\eta L D_f(x_g^k, x^*). \tag{11}$$

*Proof.* From line (7) of Algorithm 1 and optimality condition (6) it follows that

$$\|x^{k+1} - x^k\|^2 = \|\eta(y^{k+1} - y^*) + \eta(\nabla F(x_g^k) - \nabla F(x^*) - \alpha(x_g^k - x^*)) + \eta\alpha(x^{k+1} - x^*)\|^2$$
$$\geq \frac{\eta^2}{2}\|y^{k+1} - y^*\|^2 - 2\eta^2\alpha^2\|x^{k+1} - x^*\|^2$$
$$- 2\eta^2\|\nabla F(x_g^k) - \nabla F(x^*) - \alpha(x_g^k - x^*)\|^2.$$

Since $f(x) - \frac{\alpha}{2}\|x\|^2$ is a convex and $(L-\alpha)$-smooth function, we can lower bound the last term and get

$$\|x^{k+1} - x^k\|^2 = \|\eta(y^{k+1} - y^*) + \eta(\nabla F(x_g^k) - \nabla F(x^*) - \alpha(x_g^k - x^*)) + \eta\alpha(x^{k+1} - x^*)\|^2$$
$$\geq \frac{\eta^2}{2}\|y^{k+1} - y^*\|^2 - 2\eta^2\alpha^2\|x^{k+1} - x^*\|^2 - 4\eta^2(L-\alpha)D_{f - \frac{\alpha}{2}\|\cdot\|^2}(x_g^k, x^*)$$
$$\geq \frac{\eta^2}{2}\|y^{k+1} - y^*\|^2 - 2\eta^2\alpha^2\|x^{k+1} - x^*\|^2 - 4\eta^2 L D_f(x_g^k, x^*).$$

Rearranging and dividing by $2\eta$ concludes the proof. $\square$

**Lemma 3.** *Let $\mathbf{P}$ be the matrix defined by (8):*

$$\mathbf{P} = \begin{bmatrix} \frac{1}{\eta}\mathbf{I} & 0 \\ 0 & \frac{1}{\theta}\mathbf{W}^\dagger - (1+\eta\alpha)^{-1}\eta\mathbf{I} \end{bmatrix}. \tag{8}$$

*Then the following equality holds:*

$$\mathbf{P} \cdot \begin{bmatrix} x^{k+1} - x^k \\ y^{k+1} - y^k \end{bmatrix} = \begin{bmatrix} \alpha(x_g^k - x^{k+1}) - (\nabla F(x_g^k) + y^{k+1}) \\ \mathbf{W}\mathbf{W}^\dagger x^{k+1} \end{bmatrix}. \tag{12}$$

*Proof.* From the definition of $\mathbf{P}$ it follows that

$$\mathbf{P} \cdot \begin{bmatrix} x^{k+1} - x^k \\ y^{k+1} - y^k \end{bmatrix} = \begin{bmatrix} \frac{1}{\eta}(x^{k+1} - x^k) \\ \frac{1}{\theta}\mathbf{W}^\dagger(y^{k+1} - y^k) - (1+\eta\alpha)^{-1}\eta(y^{k+1} - y^k) \end{bmatrix}.$$

From line (7) of Algorithm 1 it follows that

$$\frac{1}{\eta}(x^{k+1} - x^k) = \alpha(x_g^k - x^{k+1}) - (\nabla F(x_g^k) + y^{k+1}),$$

and hence,

$$\mathbf{P} \cdot \begin{bmatrix} x^{k+1} - x^k \\ y^{k+1} - y^k \end{bmatrix} = \begin{bmatrix} \alpha(x_g^k - x^{k+1}) - (\nabla F(x_g^k) + y^{k+1}) \\ \frac{1}{\theta}\mathbf{W}^\dagger(y^{k+1} - y^k) - (1+\eta\alpha)^{-1}\eta(y^{k+1} - y^k) \end{bmatrix}.$$

From line (6) of Algorithm 1 it follows that

$$y^{k+1} - y^k = \theta\mathbf{W}x^{k+1/2},$$

and hence,

$$\mathbf{P} \cdot \begin{bmatrix} x^{k+1} - x^k \\ y^{k+1} - y^k \end{bmatrix} = \begin{bmatrix} \alpha(x_g^k - x^{k+1}) - (\nabla F(x_g^k) + y^{k+1}) \\ \mathbf{W}\mathbf{W}^\dagger x^{k+1/2} - (1+\eta\alpha)^{-1}\eta(y^{k+1} - y^k) \end{bmatrix}.$$

Since $y^k \in \mathrm{range}\mathbf{W}$ for all $k = 0, 1, 2, \ldots$, we have

$$\mathbf{W}\mathbf{W}^\dagger(y^{k+1} - y^k) = y^{k+1} - y^k,$$

and hence we obtain

$$\mathbf{P} \cdot \begin{bmatrix} x^{k+1} - x^k \\ y^{k+1} - y^k \end{bmatrix} = \begin{bmatrix} \alpha(x_g^k - x^{k+1}) - (\nabla F(x_g^k) + y^{k+1}) \\ \mathbf{W}\mathbf{W}^\dagger\left[x^{k+1/2} - (1+\eta\alpha)^{-1}\eta(y^{k+1} - y^k)\right] \end{bmatrix}.$$

Finally, from lines 5 and 7 of Algorithm 1 it follows that

$$x^{k+1} = x^{k+1/2} + (1+\eta\alpha)^{-1}\eta(y^k - y^{k+1}),$$

and hence,

$$\mathbf{P} \cdot \begin{bmatrix} x^{k+1} - x^k \\ y^{k+1} - y^k \end{bmatrix} = \begin{bmatrix} \alpha(x_g^k - x^{k+1}) - (\nabla F(x_g^k) + y^{k+1}) \\ \mathbf{W}\mathbf{W}^\dagger x^{k+1} \end{bmatrix}.$$

$\square$

**Lemma 4.** *Let parameter $\eta$ be defined by*

$$\eta = \frac{1}{4\tau L}. \tag{13}$$

*Let parameter $\theta$ be defined by*

$$\theta = \frac{1}{\eta\lambda_{\max}(\mathbf{W})}. \tag{14}$$

*Let parameter $\alpha$ be defined by*

$$\alpha = \mu. \tag{15}$$

*Let parameter $\tau$ be defined by*

$$\tau = \min\left\{1, \frac{1}{2}\sqrt{\frac{\mu}{L}\frac{\lambda_{\max}(\mathbf{W})}{\lambda_{\min}^+(\mathbf{W})}}\right\}. \tag{16}$$

*Let $\Psi^k$ be the following Lyapunov function:*

$$\Psi^k = \left\| \begin{bmatrix} x^k - x^* \\ y^k - y^* \end{bmatrix} \right\|_{\mathbf{P}}^2 + \frac{2(1-\tau)}{\tau} \mathrm{D}_F(x_f^k, x^*), \tag{17}$$

*where $\mathbf{P}$ is defined by (8):*

$$\mathbf{P} = \begin{bmatrix} \frac{1}{\eta}\mathbf{I} & 0 \\ 0 & \frac{1}{\theta}\mathbf{W}^\dagger - (1+\eta\alpha)^{-1}\eta\mathbf{I} \end{bmatrix}. \tag{8}$$

*Then the following inequality holds:*

$$\Psi^{k+1} \le \left( 1 + \frac{1}{4}\min\left\{ \sqrt{\frac{\mu}{L}\frac{\lambda_{\min}^+(\mathbf{W})}{\lambda_{\max}(\mathbf{W})}}, \frac{\lambda_{\min}^+(\mathbf{W})}{\lambda_{\max}(\mathbf{W})} \right\} \right)^{-1} \Psi^k.$$

*Proof.*

$$\left\| \begin{bmatrix} x^{k+1} - x^* \\ y^{k+1} - y^* \end{bmatrix} \right\|_{\mathbf{P}}^2 = \left\| \begin{bmatrix} x^k - x^* \\ y^k - y^* \end{bmatrix} \right\|_{\mathbf{P}}^2 - \left\| \begin{bmatrix} x^{k+1} - x^k \\ y^{k+1} - y^k \end{bmatrix} \right\|_{\mathbf{P}}^2 + 2\left\langle \mathbf{P} \cdot \begin{bmatrix} x^{k+1} - x^k \\ y^{k+1} - y^k \end{bmatrix}, \begin{bmatrix} x^{k+1} - x^* \\ y^{k+1} - y^* \end{bmatrix} \right\rangle$$

Note, that stepsize $\eta$ defined by (13) and stepsize $\theta$ defined by (14) satisfy (9), hence inequality (10) holds. Using (10) and (12) we get

$$\left\| \begin{bmatrix} x^{k+1} - x^* \\ y^{k+1} - y^* \end{bmatrix} \right\|_{\mathbf{P}}^2 \le \left\| \begin{bmatrix} x^k - x^* \\ y^k - y^* \end{bmatrix} \right\|_{\mathbf{P}}^2 - \frac{1}{\eta}\|x^{k+1} - x^k\|^2$$

$$+ 2\left\langle \begin{bmatrix} \alpha(x_g^k - x^{k+1}) - (\nabla F(x_g^k) + y^{k+1}) \\ \mathbf{W}\mathbf{W}^\dagger x^{k+1} \end{bmatrix}, \begin{bmatrix} x^{k+1} - x^* \\ y^{k+1} - y^* \end{bmatrix} \right\rangle$$

$$= \left\| \begin{bmatrix} x^k - x^* \\ y^k - y^* \end{bmatrix} \right\|_{\mathbf{P}}^2 - \frac{1}{\eta}\|x^{k+1} - x^k\|^2 + 2\alpha\langle x_g^k - x^{k+1}, x^{k+1} - x^*\rangle$$

$$- 2\langle \nabla F(x_g^k) + y^{k+1}, x^{k+1} - x^*\rangle + 2\langle \mathbf{W}\mathbf{W}^\dagger x^{k+1}, y^{k+1} - y^*\rangle.$$

Since $\mathbf{W}\mathbf{W}^\dagger x^* = 0$ and $\mathbf{W}\mathbf{W}^\dagger(y^{k+1} - y^*) = y^{k+1} - y^*$, we get

$$\left\| \begin{bmatrix} x^{k+1} - x^* \\ y^{k+1} - y^* \end{bmatrix} \right\|_{\mathbf{P}}^2 \le \left\| \begin{bmatrix} x^k - x^* \\ y^k - y^* \end{bmatrix} \right\|_{\mathbf{P}}^2 - \frac{1}{\eta}\|x^{k+1} - x^k\|^2 + 2\alpha\langle x_g^k - x^{k+1}, x^{k+1} - x^*\rangle$$

$$- 2\langle \nabla F(x_g^k) + y^{k+1}, x^{k+1} - x^*\rangle + 2\langle x^{k+1} - x^*, y^{k+1} - y^*\rangle.$$

Since $\nabla F(x^*) + y^* = 0$ (optimality condition (6)), we get

$$\left\| \begin{bmatrix} x^{k+1} - x^* \\ y^{k+1} - y^* \end{bmatrix} \right\|_{\mathbf{P}}^2 \le \left\| \begin{bmatrix} x^k - x^* \\ y^k - y^* \end{bmatrix} \right\|_{\mathbf{P}}^2 - \frac{1}{\eta}\|x^{k+1} - x^k\|^2 + 2\alpha\langle x_g^k - x^{k+1}, x^{k+1} - x^*\rangle$$
$$- 2\langle \nabla F(x_g^k) - \nabla F(x^*) + y^{k+1} - y^*, x^{k+1} - x^*\rangle + 2\langle x^{k+1} - x^*, y^{k+1} - y^*\rangle$$

$$= \left\| \begin{bmatrix} x^k - x^* \\ y^k - y^* \end{bmatrix} \right\|_{\mathbf{P}}^2 - \frac{1}{\eta}\|x^{k+1} - x^k\|^2 - 2\alpha\|x^{k+1} - x^*\|^2$$
$$- 2\alpha\langle x_g^k - x^*, x^{k+1} - x^*\rangle - 2\langle \nabla F(x_g^k) - \nabla F(x^*), x^{k+1} - x^*\rangle.$$

Using Young's inequality $2\langle a, b\rangle \le \|a\|^2 + \|b\|^2$ we get

$$\left\| \begin{bmatrix} x^{k+1} - x^* \\ y^{k+1} - y^* \end{bmatrix} \right\|_{\mathbf{P}}^2 \le \left\| \begin{bmatrix} x^k - x^* \\ y^k - y^* \end{bmatrix} \right\|_{\mathbf{P}}^2 - \frac{1}{\eta}\|x^{k+1} - x^k\|^2 - 2\alpha\|x^{k+1} - x^*\|^2$$
$$+ \alpha\|x_g^k - x^*\|^2 + \alpha\|x^{k+1} - x^*\|^2 - 2\langle \nabla F(x_g^k) - \nabla F(x^*), x^{k+1} - x^*\rangle$$

$$= \left\| \begin{bmatrix} x^k - x^* \\ y^k - y^* \end{bmatrix} \right\|_{\mathbf{P}}^2 - \frac{1}{\eta}\|x^{k+1} - x^k\|^2 - \alpha\|x^{k+1} - x^*\|^2 + \alpha\|x_g^k - x^*\|^2$$
$$- 2\langle \nabla F(x_g^k) - \nabla F(x^*), x^{k+1} - x^*\rangle.$$

Now, we use lines 4 and 8 of Algorithm 1 and get

$$\left\| \begin{bmatrix} x^{k+1} - x^* \\ y^{k+1} - y^* \end{bmatrix} \right\|_{\mathbf{P}}^2 \le \left\| \begin{bmatrix} x^k - x^* \\ y^k - y^* \end{bmatrix} \right\|_{\mathbf{P}}^2 - \alpha\|x^{k+1} - x^*\|^2 + \alpha\|x_g^k - x^*\|^2 - \frac{1}{2\eta}\|x^{k+1} - x^k\|^2$$
$$- \frac{2-\tau}{\tau}\left( \langle \nabla F(x_g^k) - \nabla F(x^*), x_f^{k+1} - x_g^k\rangle + \frac{1}{2\eta}\frac{(2-\tau)}{4\tau}\|x_f^{k+1} - x_g^k\|^2 \right)$$
$$- 2\langle \nabla F(x_g^k) - \nabla F(x^*), x_g^k - x^*\rangle + \frac{2(1-\tau)}{\tau}\langle \nabla F(x_g^k) - \nabla F(x^*), x_f^k - x_g^k\rangle.$$

Since parameter $\eta$ defined by (13) satisfy $\eta \le \frac{2-\tau}{4\tau L}$, we get

$$\left\| \begin{bmatrix} x^{k+1} - x^* \\ y^{k+1} - y^* \end{bmatrix} \right\|_{\mathbf{P}}^2 \le \left\| \begin{bmatrix} x^k - x^* \\ y^k - y^* \end{bmatrix} \right\|_{\mathbf{P}}^2 - \alpha\|x^{k+1} - x^*\|^2 + \alpha\|x_g^k - x^*\|^2 - \frac{1}{2\eta}\|x^{k+1} - x^k\|^2$$
$$- \frac{2-\tau}{\tau}\left( \langle \nabla F(x_g^k) - \nabla F(x^*), x_f^{k+1} - x_g^k\rangle + \frac{L}{2}\|x_f^{k+1} - x_g^k\|^2 \right)$$
$$- 2\langle \nabla F(x_g^k) - \nabla F(x^*), x_g^k - x^*\rangle + \frac{2(1-\tau)}{\tau}\langle \nabla F(x_g^k) - \nabla F(x^*), x_f^k - x_g^k\rangle.$$

Using $\mu$-strong convexity and $L$-smoothness of $f(x)$ we get

$$\left\| \begin{bmatrix} x^{k+1} - x^* \\ y^{k+1} - y^* \end{bmatrix} \right\|_{\mathbf{P}}^2 \le \left\| \begin{bmatrix} x^k - x^* \\ y^k - y^* \end{bmatrix} \right\|_{\mathbf{P}}^2 - \alpha\|x^{k+1} - x^*\|^2 + \alpha\|x_g^k - x^*\|^2 - \frac{1}{2\eta}\|x^{k+1} - x^k\|^2$$
$$- \frac{2-\tau}{\tau}\left( \mathrm{D}_F(x_f^{k+1}, x^*) - \mathrm{D}_F(x_g^k, x^*) \right) + \frac{2(1-\tau)}{\tau}\left( \mathrm{D}_F(x_f^k, x^*) - \mathrm{D}_F(x_g^k, x^*) \right)$$

$$- 2\left(\mathrm{D}_F(x_g^k, x^*) + \frac{\mu}{2}\|x_g^k - x^*\|^2\right)$$

$$= \left\|\begin{bmatrix} x^k - x^* \\ y^k - y^* \end{bmatrix}\right\|_{\mathbf{P}}^2 - \alpha\|x^{k+1} - x^*\|^2 + \frac{2(1-\tau)}{\tau}\mathrm{D}_F(x_f^k, x^*) - \frac{2-\tau}{\tau}\mathrm{D}_F(x_f^{k+1}, x^*)$$

$$+ (\alpha - \mu)\|x_g^k - x^*\|^2 - \frac{1}{2\eta}\|x^{k+1} - x^k\|^2 - \mathrm{D}_F(x_g^k, x^*).$$

Now, we define $\delta = \min\left\{1, \frac{1}{2\eta L}\right\}$. Since $\alpha$ defined by (15) satisfies conditions of Lemma 2, we can use (11) and get

$$\left\|\begin{bmatrix} x^{k+1} - x^* \\ y^{k+1} - y^* \end{bmatrix}\right\|_{\mathbf{P}}^2 \leq \left\|\begin{bmatrix} x^k - x^* \\ y^k - y^* \end{bmatrix}\right\|_{\mathbf{P}}^2 - \alpha\|x^{k+1} - x^*\|^2 + \frac{2(1-\tau)}{\tau}\mathrm{D}_F(x_f^k, x^*) - \frac{2-\tau}{\tau}\mathrm{D}_F(x_f^{k+1}, x^*)$$

$$+ (\alpha - \mu)\|x_g^k - x^*\|^2 - \frac{\delta}{2\eta}\|x^{k+1} - x^k\|^2 - \mathrm{D}_F(x_g^k, x^*)$$

$$\leq \left\|\begin{bmatrix} x^k - x^* \\ y^k - y^* \end{bmatrix}\right\|_{\mathbf{P}}^2 - \alpha\|x^{k+1} - x^*\|^2 + \frac{2(1-\tau)}{\tau}\mathrm{D}_F(x_f^k, x^*) - \frac{2-\tau}{\tau}\mathrm{D}_F(x_f^{k+1}, x^*)$$

$$- \frac{\eta\delta}{4}\|y^{k+1} - y^*\|^2 + \eta\alpha^2\delta\|x^{k+1} - x^*\|^2 + 2\eta L\delta\mathrm{D}_f(x_g^k, x^*)$$

$$+ (\alpha - \mu)\|x_g^k - x^*\|^2 - \mathrm{D}_F(x_g^k, x^*)$$

$$\leq \left\|\begin{bmatrix} x^k - x^* \\ y^k - y^* \end{bmatrix}\right\|_{\mathbf{P}}^2 - \alpha\|x^{k+1} - x^*\|^2 + \frac{2(1-\tau)}{\tau}\mathrm{D}_F(x_f^k, x^*) - \frac{2-\tau}{\tau}\mathrm{D}_F(x_f^{k+1}, x^*)$$

$$- \frac{\eta\delta}{4}\|y^{k+1} - y^*\|^2 + \frac{\alpha^2}{2L}\|x^{k+1} - x^*\|^2 + (\alpha - \mu)\|x_g^k - x^*\|^2$$

$$= \left\|\begin{bmatrix} x^k - x^* \\ y^k - y^* \end{bmatrix}\right\|_{\mathbf{P}}^2 - \left(\alpha - \frac{\alpha^2}{2L}\right)\|x^{k+1} - x^*\|^2 - \frac{\eta\delta}{4}\|y^{k+1} - y^*\|^2$$

$$+ \frac{2(1-\tau)}{\tau}\mathrm{D}_F(x_f^k, x^*) - \frac{2-\tau}{\tau}\mathrm{D}_F(x_f^{k+1}, x^*) + (\alpha - \mu)\|x_g^k - x^*\|^2.$$

Using parameter $\alpha$ defined by (15) we get

$$\left\|\begin{bmatrix} x^{k+1} - x^* \\ y^{k+1} - y^* \end{bmatrix}\right\|_{\mathbf{P}}^2 \leq \left\|\begin{bmatrix} x^k - x^* \\ y^k - y^* \end{bmatrix}\right\|_{\mathbf{P}}^2 - \frac{\mu}{2}\|x^{k+1} - x^*\|^2 - \frac{\eta\delta}{4}\|y^{k+1} - y^*\|^2$$

$$+ \frac{2(1-\tau)}{\tau}\mathrm{D}_F(x_f^k, x^*) - \frac{2-\tau}{\tau}\mathrm{D}_F(x_f^{k+1}, x^*).$$

Since $y^k, y^* \in \mathrm{range}\mathbf{W}$, we get

$$\left\|\begin{bmatrix} x^{k+1} - x^* \\ y^{k+1} - y^* \end{bmatrix}\right\|_{\mathbf{P}}^2 \leq \left\|\begin{bmatrix} x^k - x^* \\ y^k - y^* \end{bmatrix}\right\|_{\mathbf{P}}^2 - \frac{\mu}{2}\|x^{k+1} - x^*\|^2 - \frac{\eta\delta\lambda_{\min}^+(\mathbf{W})}{4}\|y^{k+1} - y^*\|_{\mathbf{W}^\dagger}^2$$

$$+ \frac{2(1-\tau)}{\tau}\mathrm{D}_F(x_f^k, x^*) - \frac{2-\tau}{\tau}\mathrm{D}_F(x_f^{k+1}, x^*).$$

Using (10) we get

$$\left\| \begin{bmatrix} x^{k+1} - x^* \\ y^{k+1} - y^* \end{bmatrix} \right\|_{\mathbf{P}}^2 \leq \left\| \begin{bmatrix} x^k - x^* \\ y^k - y^* \end{bmatrix} \right\|_{\mathbf{P}}^2 - \min\left\{ \frac{\eta\mu}{2}, \frac{\eta\theta\delta\lambda_{\min}^+(\mathbf{W})}{4} \right\} \left\| \begin{bmatrix} x^{k+1} - x^* \\ y^{k+1} - y^* \end{bmatrix} \right\|_{\mathbf{P}}^2$$
$$+ \frac{2(1-\tau)}{\tau} \mathrm{D}_F(x_f^k, x^*) - \frac{2-\tau}{\tau} \mathrm{D}_F(x_f^{k+1}, x^*).$$

Using parameter $\theta$ defined by (14) and definition of $\delta$ we get

$$\left\| \begin{bmatrix} x^{k+1} - x^* \\ y^{k+1} - y^* \end{bmatrix} \right\|_{\mathbf{P}}^2 \leq \left\| \begin{bmatrix} x^k - x^* \\ y^k - y^* \end{bmatrix} \right\|_{\mathbf{P}}^2 - \min\left\{ \frac{\eta\mu}{2}, \frac{\lambda_{\min}^+(\mathbf{W})}{4\lambda_{\max}(\mathbf{W})}, \frac{\lambda_{\min}^+(\mathbf{W})}{8\eta L\lambda_{\max}(\mathbf{W})} \right\} \left\| \begin{bmatrix} x^{k+1} - x^* \\ y^{k+1} - y^* \end{bmatrix} \right\|_{\mathbf{P}}^2$$
$$+ \frac{2(1-\tau)}{\tau} \mathrm{D}_F(x_f^k, x^*) - \frac{2-\tau}{\tau} \mathrm{D}_F(x_f^{k+1}, x^*).$$

Plugging parameter $\eta$ defined by (13) we get

$$\left\| \begin{bmatrix} x^{k+1} - x^* \\ y^{k+1} - y^* \end{bmatrix} \right\|_{\mathbf{P}}^2 \leq \left\| \begin{bmatrix} x^k - x^* \\ y^k - y^* \end{bmatrix} \right\|_{\mathbf{P}}^2 - \min\left\{ \frac{\mu}{8\tau L}, \frac{\lambda_{\min}^+(\mathbf{W})}{4\lambda_{\max}(\mathbf{W})}, \frac{\tau\lambda_{\min}^+(\mathbf{W})}{2\lambda_{\max}(\mathbf{W})} \right\} \left\| \begin{bmatrix} x^{k+1} - x^* \\ y^{k+1} - y^* \end{bmatrix} \right\|_{\mathbf{P}}^2$$
$$+ \frac{2(1-\tau)}{\tau} \mathrm{D}_F(x_f^k, x^*) - \frac{2-\tau}{\tau} \mathrm{D}_F(x_f^{k+1}, x^*)$$
$$\leq \left\| \begin{bmatrix} x^k - x^* \\ y^k - y^* \end{bmatrix} \right\|_{\mathbf{P}}^2 - \min\left\{ \frac{\mu}{8\tau L}, \frac{\lambda_{\min}^+(\mathbf{W})}{4\lambda_{\max}(\mathbf{W})}, \frac{\tau\lambda_{\min}^+(\mathbf{W})}{2\lambda_{\max}(\mathbf{W})} \right\} \left\| \begin{bmatrix} x^{k+1} - x^* \\ y^{k+1} - y^* \end{bmatrix} \right\|_{\mathbf{P}}^2$$
$$+ \frac{2(1-\tau)}{\tau} \mathrm{D}_F(x_f^k, x^*) - \left(1 + \frac{\tau}{2}\right)\frac{2(1-\tau)}{\tau} \mathrm{D}_F(x_f^{k+1}, x^*).$$

After rearranging and using definition of $\Psi^k$ (17) we get

$$\Psi^k \geq \left(1 + \min\left\{ \frac{\tau}{2}, \frac{\mu}{8\tau L}, \frac{\lambda_{\min}^+(\mathbf{W})}{4\lambda_{\max}(\mathbf{W})}, \frac{\tau\lambda_{\min}^+(\mathbf{W})}{2\lambda_{\max}(\mathbf{W})} \right\}\right) \Psi^{k+1}.$$

Plugging parameter $\tau$ defined by (16) we get

$$\Psi^k \geq \left(1 + \frac{1}{4}\min\left\{ \sqrt{\frac{\mu}{L}\frac{\lambda_{\min}^+(\mathbf{W})}{\lambda_{\max}(\mathbf{W})}}, \frac{\lambda_{\min}^+(\mathbf{W})}{\lambda_{\max}(\mathbf{W})} \right\}\right) \Psi^{k+1}.$$

$\square$

*Proof of Theorem 2 (APAPC).* Conditions of Lemma 4 are satisfied, hence the following inequality holds for all $k$:

$$\Psi^{k+1} \leq \left(1 + \frac{1}{4}\min\left\{ \sqrt{\frac{\mu}{L}\frac{\lambda_{\min}^+(\mathbf{W})}{\lambda_{\max}(\mathbf{W})}}, \frac{\lambda_{\min}^+(\mathbf{W})}{\lambda_{\max}(\mathbf{W})} \right\}\right)^{-1} \Psi^k.$$

After doing telescoping we get

$$\Psi^k \leq \left(1 + \frac{1}{4}\min\left\{ \sqrt{\frac{\mu}{L}\frac{\lambda_{\min}^+(\mathbf{W})}{\lambda_{\max}(\mathbf{W})}}, \frac{\lambda_{\min}^+(\mathbf{W})}{\lambda_{\max}(\mathbf{W})} \right\}\right)^{-k} \Psi^0.$$

Inequality (10) implies $\Psi^0 \leq C$, where $C := \frac{1}{\eta}\left\| x^0 - x^* \right\|^2 + \frac{1}{\theta}\|y^0 - y^*\|_{\mathbf{W}^\dagger}^2 + \frac{2(1-\tau)}{\tau}\mathrm{D}_F(x_f^0, x^*)$. Hence, we obtain.

$$\Psi^k \leq \left(1 + \frac{1}{4}\min\left\{ \sqrt{\frac{\mu}{L}\frac{\lambda_{\min}^+(\mathbf{W})}{\lambda_{\max}(\mathbf{W})}}, \frac{\lambda_{\min}^+(\mathbf{W})}{\lambda_{\max}(\mathbf{W})} \right\}\right)^{-k} C.$$

It remains to lower bound $\Psi^k$ using (10) one more time:

$$\frac{1}{\eta}\left\|x^k - x^*\right\|^2 + \frac{2(1-\tau)}{\tau}D_F(x_f^k, x^*) \leq \Psi^k \leq \left(1 + \frac{1}{4}\min\left\{\sqrt{\frac{\mu}{L}\frac{\lambda_{\min}^+(\mathbf{W})}{\lambda_{\max}(\mathbf{W})}}, \frac{\lambda_{\min}^+(\mathbf{W})}{\lambda_{\max}(\mathbf{W})}\right\}\right)^{-k} C.$$

Finally, choosing number of iterations

$$k \geq \left(1 + 4\max\left\{\sqrt{\frac{L\lambda_{\max}(\mathbf{W})}{\mu\lambda_{\min}^+(\mathbf{W})}}, \frac{\lambda_{\max}(\mathbf{W})}{\lambda_{\min}^+(\mathbf{W})}\right\}\right)\log\left(\frac{\eta C}{\varepsilon}\right).$$

implies $\|x^k - x^*\|^2 \leq \varepsilon$. $\qquad\square$

# D  Proof of Corollary 1 (OPAPC)

First, Theorem 2 still holds true by replacing $\lambda_{\max}(\mathbf{W})$ by an upper bound $\lambda_1$, $\lambda_{\min}^+(\mathbf{W})$ by a lower bound $\lambda_2 > 0$ and $\chi(\mathbf{W})$ by the upper bound $\chi = \lambda_1/\lambda_2$.[2]

The proof of Corollary 1 is similar to the proof of Theorem 4 of Scaman et al. (2017).

Denote $\tilde{\mathbf{W}} = \frac{2\chi(\mathbf{W})}{(1+\chi(\mathbf{W}))\lambda_{\max}(\mathbf{W})}\mathbf{W}$. Let $I$ be the interval $I = [1 - \frac{1}{c_2}, 1 + \frac{1}{c_2}] \subset (0, 2)$. Then, $\mathrm{Sp}(\tilde{\mathbf{W}}) \setminus \{0\} \subset I$, where $\mathrm{Sp}$ denotes the spectrum. Moreover, using Scaman et al. (2017), the polynomial $P$ satisfies $P(0) = 0$ and $\max_{t \in I} |1 - P(t)| = \frac{2c_1^T}{1 + c_1^{2T}} < 1$. Therefore,

$$\mathrm{Sp}(I - P(\tilde{\mathbf{W}})) \setminus \{1\} \subset \left[ -\frac{2c_1^T}{1 + c_1^{2T}}, \frac{2c_1^T}{1 + c_1^{2T}} \right] \subset (-1, 1).$$

Consequently,

$$\lambda_{\max}(P(\tilde{\mathbf{W}})) \leq \lambda_1 := 1 + \frac{2c_1^T}{1 + c_1^{2T}} < 2, \quad \lambda_{\min}^+(P(\tilde{\mathbf{W}})) \geq \lambda_2 := 1 - \frac{2c_1^T}{1 + c_1^{2T}} > 0.$$

Moreover, by replacing $c_1$ and $T$ by their values, $\chi := \frac{\lambda_1}{\lambda_2} \leq 4$, see (Scaman et al., 2017, Equation 34).

Applying APAPC with the gossip matrix $P(\tilde{\mathbf{W}})$ leads to OPAPC. Then, we apply Theorem 2 to OPAPC. More precisely, we apply Theorem 2 by replacing $\mathbf{W}$ by $P(\tilde{\mathbf{W}})$ and $\lambda_{\max}(\mathbf{W})$ (resp. $\lambda_{\min}^+(\mathbf{W})$) by the upper bound (resp. the lower bound) $\lambda_1$ (resp. $\lambda_2$) of $\lambda_{\max}(P(\tilde{\mathbf{W}}))$ (resp. $\lambda_{\min}^+(P(\tilde{\mathbf{W}}))$). Denoting $x^k$ the iterates of OPAPC, we obtain

$$\frac{1}{\eta} \left\| x^k - x^* \right\|^2 + \frac{2 - \tau}{\tau} D_F(x_f^k, x^*) \leq \left( 1 + \frac{1}{16} \min\left\{ \frac{2}{\sqrt{\kappa}}, 1 \right\} \right)^{-k} C.$$

Finally, the gradient computation complexity of OPAPC is $\mathcal{O}(\sqrt{\kappa} \log(1/\varepsilon))$. One multiplication by $P(\tilde{\mathbf{W}})$ is equivalent to one application of the procedure ACCELERATEDGOSSIP$(\mathbf{W}, \cdot, T)$, which requires exactly $T$ communication rounds. Therefore, the communication complexity of OPAPC is $T\mathcal{O}(\sqrt{\kappa} \log(1/\varepsilon)) = \mathcal{O}(\sqrt{\kappa \chi(\mathbf{W})} \log(1/\varepsilon))$.

# E   A Loopless Algorithm Optimal in Communication Complexity

We propose another accelerated Forward Backward algorithm to solve Problem (3). More precisely, we first provide a reformulation of Problem (3), different from the reformulation (6). Then, we design an accelerated Forward Backward algorithm associated with this reformulation. Remarkably, the matrix $\mathbf{W}$ is only involved in the operator $A$ of this new Forward Backward algorithm. This leads to an acceleration compared to APAPC, and to an optimal communication complexity.

In this section, $\mathsf{E}$ is the Euclidean space $\mathsf{E} = (\mathbb{R}^d)^{\mathcal{V}} \times (\mathbb{R}^d)^{\mathcal{V}} \times \mathrm{range}(\mathbf{W})$ endowed with the norm $\|(x, y, z)\|_{\mathsf{E}}^2 := \|x\|^2 + \|y\|^2 + \|z\|_{\mathbf{W}^\dagger}^2$.

Using the first order optimality conditions, a point $x^*$ is a solution to Problem (3) if and only if $\nabla F(x^*) \in \mathrm{range}(\mathbf{W})$ and $x^* \in \ker(\mathbf{W})$. Solving Problem (3) is therefore equivalent to finding $(x^*, y^*, z^*) \in \mathsf{E}$ such that

$$0 = \nabla F(x^*) - \frac{\mu}{2}x^* - y^*, \tag{18}$$

$$0 = x^* + \frac{2}{\mu}(y^* + z^*), \tag{19}$$

$$0 = \frac{2}{\mu}\mathbf{W}(y^* + z^*). \tag{20}$$

Indeed, if (18)–(20) holds, then using (18), $y^* = \nabla F(x^*) - \frac{\mu}{2}x^*$ and using (19) $z^* = -\nabla F(x^*) \in \mathrm{range}(\mathbf{W})$. Since $z^* \in \mathrm{range}(\mathbf{W})$ and $y^* + z^* = -\frac{\mu}{2}x^*$, we have $\nabla F(x^*) \in \mathrm{range}(\mathbf{W})$ and $x^* \in \ker(\mathbf{W})$. On the other hand, if $\nabla F(x^*) \in \mathrm{range}(\mathbf{W})$ and $x^* \in \ker(\mathbf{W})$, then $\mathbf{W}x^* = 0$ and setting $y^* = \nabla F(x^*) - \frac{\mu}{2}x^*$ and $z^* = -\nabla F(x^*) \in \mathrm{range}(\mathbf{W})$ leads to (18)–(20).

Consider the map $M : \mathsf{E} \to \mathsf{E}$

$$M(x, y, z) := \begin{bmatrix} \nabla F(x) - \frac{\mu}{2}x & -y \\ x & +\frac{2}{\mu}y & +\frac{2}{\mu}z \\ & \frac{2}{\mu}\mathbf{W}y & +\frac{2}{\mu}\mathbf{W}z \end{bmatrix}.$$

Similary to Section 4.1, one can show that $M$ is a monotone operator. Moreover, $M(x^*, y^*, z^*) = 0$, i.e., $(x^*, y^*, z^*)$ is a zero of $M$.

Consider the maps $A, B : \mathsf{E} \to \mathsf{E}$ defined by

$$A(x, y, z) = \begin{bmatrix} \nabla F(x) - \frac{\mu}{2}x \\ \frac{2}{\mu}(y + z) + \nu y \\ \frac{2}{\mu}\mathbf{W}(y + z) \end{bmatrix}, \quad B(x, y, z) = \begin{bmatrix} -y \\ x - \nu y \\ 0 \end{bmatrix}.$$

Then, $M = A + B$. Note that there is a term $\nu y$, where $\nu > 0$ in $A(x, y, z)$ and a term $-\nu y$ in $B(x, y, z)$, which cancel out in the sum $A(x, y, z) + B(x, y, z)$. This additional term makes the operator $A(x, y, z)$ strongly monotone. Indeed, $A$ is the gradient of the strongly convex function (in $\mathsf{E}$) $\mathsf{E} \ni (x, y, z) \mapsto r(x) + h(y, z)$ defined by

$$r(x) := F(x) - \frac{\mu}{4}\|x\|^2, \quad h(y, z) := \frac{1}{\mu}\|y + z\|^2 + \frac{\nu}{2}\|y\|^2.$$

In other words, operator $A(x, y, z)$ can be written as

$$A(x, y, z) = \begin{bmatrix} \nabla r(x) \\ \nabla_y h(y, z) \\ \mathbf{W}\nabla_z h(y, z) \end{bmatrix},$$

and one can check that $A$ is strongly monotone. However, the operator $B(x, y, z)$ is not monotone in general. Indeed, $B$ is only weakly monotone since $B$ satisfies

$$\left\langle B(x, y, z) - B(x^*, y^*, z^*), \begin{bmatrix} x - x^* \\ y - y^* \\ z - z^* \end{bmatrix}_E \right\rangle = -\nu \|y - y^*\|^2.$$

One idea to solve (18)–(20) is to apply Algorithm (4) to the sum $A + B$, although $B$ is not monotone. Note that $B$ is linear and, although $B$ is not monotone, the resolvent of $B$ is still well defined while $1 - \gamma\nu + \gamma^2 \neq 0$. Indeed, $(x', y') = J_{\gamma B}(x, y)$ implies $x' = x + \gamma y'$, and $(1 - \gamma\nu + \gamma^2)y' = y - \gamma x$.

In particular, we propose a new algorithm that can be seen as an accelerated version of the Forward Backward Algorithm (4) to find a zero of $A + B$. The proposed algorithm is defined in Algorithm 3 and its complexity is given in Theorem 3. We show that the complexity of Algorithm 3 is $\mathcal{O}(\sqrt{\kappa\chi(\mathbf{W})}\log(1/\varepsilon))$, both in communication rounds and gradient computations. The proposed algorithm is therefore optimal in communication complexity, see Section 3.2. Moreover, Algorithm 3 uses only one gradient computation by communication round.

---

**Algorithm 3**

---

1: **Parameters:** $x^0, y^0 \in \mathbb{R}^{nd}, z^0 \in \mathrm{range}\mathbf{W}, \eta, \theta, \lambda, \alpha, \beta, \gamma, \nu > 0, \tau, \sigma \in (0, 1)$
2: Set $x_f^0 = x^0$
3: Set $y_f^0 = y^0$
4: Set $z_f^0 = z^0$
5: **for** $k = 0, 1, 2, \ldots$ **do**
6: $\quad x_g^k = \tau x^k + (1 - \tau)x_f^k$
7: $\quad y_g^k = \sigma y^k + (1 - \sigma)y_f^k$
8: $\quad z_g^k = \sigma z^k + (1 - \sigma)z_f^k$
9: $\quad x^{k+1} = x^k + \eta\alpha(x_g^k - x^{k+1}) - \eta\nabla r(x_g^k) + \eta y^{k+1}$
10: $\quad y^{k+1} = y^k + \theta\beta(y_g^k - y^{k+1}) - \theta\nabla_y h(y_g^k, z_g^k) + \theta\nu y^{k+1} - \theta x^{k+1}$
11: $\quad z^{k+1} = z^k + \lambda\gamma(z_g^k - z^{k+1}) - \lambda\mathbf{W}\nabla_z h(y_g^k, z_g^k)$
12: $\quad x_f^{k+1} = x_g^k + \frac{2\tau}{2-\tau}(x^{k+1} - x^k)$
13: $\quad y_f^{k+1} = y_g^k + \sigma(y^{k+1} - y^k)$
14: $\quad z_f^{k+1} = z_g^k + \sigma(z^{k+1} - z^k)$
15: **end for**

---

**Theorem 3** (Algorithm 3). *Set the parameters $\eta, \theta, \lambda, \alpha, \beta, \gamma, \nu > 0, \tau, \sigma \in (0, 1)$ to*

$$\eta = \left[2\sqrt{L\mu} + \mu\right]^{-1}, \qquad\qquad \alpha = \frac{\mu}{3}, \qquad \tau = \frac{1}{2}\sqrt{\frac{\mu}{L}},$$

$$\theta = \left[\frac{1}{4}\sqrt{\frac{\lambda_{\min}^+(\mathbf{W})}{\lambda_{\max}(\mathbf{W})\mu L}} + \frac{5}{96L}\right]^{-1}, \qquad \beta = \frac{1}{96L}, \qquad \sigma = \frac{1}{20}\sqrt{\frac{\lambda_{\min}^+(\mathbf{W})}{\lambda_{\max}(\mathbf{W})}\frac{\mu}{L}},$$

$$\lambda = \left[\frac{1}{4}\sqrt{\frac{\lambda_{\min}^+(\mathbf{W})\lambda_{\max}(\mathbf{W})}{\mu L}} + \frac{\lambda_{\min}^+(\mathbf{W})}{96L}\right]^{-1}, \quad \gamma = \frac{\lambda_{\min}^+(\mathbf{W})}{96L}, \quad \nu = \frac{1}{24L}.$$

*Then, the sequence $(x^k)$ converges linearly to $x^*$. Moreover, for every $\varepsilon > 0$, Algorithm 3 finds $x^k$ for which $\|x^k - x^*\|^2 \leq \varepsilon$ in at most $\mathcal{O}\left(\sqrt{\kappa\chi(\mathbf{W})}\log(1/\varepsilon)\right)$ gradient computations (resp. communication rounds).*

The Algorithm 3 achieves the communication lower bound of Theorem 1. The proof of Theorem 3 intuitively relies on viewing Algorithm 3 as an accelerated version of (4), although Nesterov's acceleration does not apply to general monotone operators and even less to non monotone operators.

# F    Proof of Theorem 3 (Algorithm 3)

**Lemma 5.** *Let $\alpha$ satisfy*

$$\alpha \leq \frac{\mu}{2}. \tag{21}$$

*Let $\delta$ be defined by*

$$\delta = \min\left\{1, \frac{1}{2\eta L}\right\}. \tag{22}$$

*Then the following inequality holds:*

$$-\frac{1}{2\eta}\|x^{k+1} - x^*\|^2 \leq -\frac{\eta\delta}{4}\|y^{k+1} - y^*\|^2 + \frac{\alpha}{4}\|x^{k+1} - x^*\|^2 + D_r(x_g^k, x^*).$$

*Proof.* From line 9 of Algortihm 3 it follows that

$$x^{k+1} - x^k = \eta\alpha(x_g^k - x^{k+1}) - \eta\nabla r(x_g^k) + \eta y^{k+1}.$$

From optimality condition (18) it follows that $\nabla r(x^*) = y^*$ and hence

$$\|x^{k+1} - x^k\|^2 = \eta^2\|y^{k+1} - y^* - \alpha(x^{k+1} - x^*) - (\nabla r(x_g^k) - \nabla r(x^*) - \alpha(x_g^k - x^*))\|^2$$

$$\geq \frac{\eta^2}{2}\|y^{k+1} - y^*\|^2 - \eta^2\|\alpha(x^{k+1} - x^*) + (\nabla r(x_g^k) - \nabla r(x^*) - \alpha(x_g^k - x^*))\|^2$$

$$\geq \frac{\eta^2}{2}\|y^{k+1} - y^*\|^2 - 2\eta^2\alpha^2\|x^{k+1} - x^*\|^2$$

$$- 2\eta^2\|\nabla r(x_g^k) - \nabla r(x^*) - \alpha(x_g^k - x^*)\|^2.$$

From (21) it follows that function $r(x) - \frac{\alpha}{2}\|x\|^2 = F(x) - \frac{\mu+2\alpha}{4}\|x\|^2$ is convex and $L$-smooth, hence we can bound the last term:

$$\|x^{k+1} - x^k\|^2 \geq \frac{\eta^2}{2}\|y^{k+1} - y^*\|^2 - 2\eta^2\alpha^2\|x^{k+1} - x^*\|^2 - 4\eta^2 L D_{r(\cdot) - \frac{\alpha}{2}\|\cdot\|^2}(x_g^k, x^*)$$

$$= \frac{\eta^2}{2}\|y^{k+1} - y^*\|^2 - 2\eta^2\alpha^2\|x^{k+1} - x^*\|^2 - 4\eta^2 L D_r(x_g^k, x^*) + 2\eta^2 L\alpha\|x_g^k - x^*\|^2$$

$$\geq \frac{\eta^2}{2}\|y^{k+1} - y^*\|^2 - 2\eta^2\alpha^2\|x^{k+1} - x^*\|^2 - 4\eta^2 L D_r(x_g^k, x^*).$$

Multiplying by $\frac{1}{2\eta}$ and rearranging gives

$$-\frac{1}{2\eta}\|x^{k+1} - x^*\|^2 \leq -\frac{\eta}{4}\|y^{k+1} - y^*\|^2 + \eta\alpha^2\|x^{k+1} - x^*\|^2 + 2\eta L D_r(x_g^k, x^*).$$

Using $\delta$ defined by (22) we obtain

$$-\frac{1}{2\eta}\|x^{k+1} - x^*\|^2 \leq -\frac{\delta}{2\eta}\|x^{k+1} - x^*\|^2$$

$$\leq -\frac{\eta\delta}{4}\|y^{k+1} - y^*\|^2 + \delta\eta\alpha^2\|x^{k+1} - x^*\|^2 + 2\delta\eta L D_r(x_g^k, x^*)$$

$$\leq -\frac{\eta\delta}{4}\|y^{k+1} - y^*\|^2 + \frac{\eta\alpha^2}{2\eta L}\|x^{k+1} - x^*\|^2 + D_r(x_g^k, x^*)$$

$$\leq -\frac{\eta\delta}{4}\|y^{k+1} - y^*\|^2 + \frac{\alpha\mu}{4L}\|x^{k+1} - x^*\|^2 + D_r(x_g^k, x^*)$$

$$\leq -\frac{\eta\delta}{4}\|y^{k+1} - y^*\|^2 + \frac{\alpha}{4}\|x^{k+1} - x^*\|^2 + D_r(x_g^k, x^*).$$

$\square$

**Lemma 6.** *Let $\alpha$ satisfy*

$$\alpha \leq \frac{\mu}{2}. \tag{21}$$

*Let $\eta$ satisfy*

$$\eta \le \frac{1}{4\tau L}. \tag{23}$$

*Then the following inequality holds:*

$$\frac{1}{\eta}\|x^{k+1} - x^*\|^2 \le \frac{1}{\eta}\|x^k - x^*\|^2 - \frac{3\alpha}{4}\|x^{k+1} - x^*\|^2 + \frac{2(1-\tau)}{\tau}\mathrm{D}_r(x_f^k, x^*) - \frac{2-\tau}{\tau}\mathrm{D}_r(x_f^{k+1}, x^*) \tag{24}$$

$$- \frac{\eta\delta}{4}\|y^{k+1} - y^*\|^2 + 2\langle y^{k+1} - y^*, x^{k+1} - x^*\rangle.$$

*Proof.* Using line 9 of Algorithm 3 we get

$$\frac{1}{\eta}\|x^{k+1} - x^*\|^2 = \frac{1}{\eta}\|x^k - x^*\|^2 + \frac{2}{\eta}\langle x^{k+1} - x^k, x^{k+1} - x^*\rangle - \frac{1}{\eta}\|x^{k+1} - x^k\|^2$$

$$= \frac{1}{\eta}\|x^k - x^*\|^2 - \frac{1}{\eta}\|x^{k+1} - x^k\|^2 + 2\alpha\langle x_g^k - x^{k+1}, x^{k+1} - x^*\rangle$$

$$- 2\langle \nabla r(x_g^k) - y^{k+1}, x^{k+1} - x^*\rangle$$

$$= \frac{1}{\eta}\|x^k - x^*\|^2 - \frac{1}{\eta}\|x^{k+1} - x^k\|^2 + 2\alpha\langle x_g^k - x^*, x^{k+1} - x^*\rangle - 2\alpha\|x^{k+1} - x^*\|^2$$

$$- 2\langle \nabla r(x_g^k) - y^{k+1}, x^{k+1} - x^*\rangle$$

$$\le \frac{1}{\eta}\|x^k - x^*\|^2 - \frac{1}{\eta}\|x^{k+1} - x^k\|^2 + \alpha\|x_g^k - x^*\|^2 - \alpha\|x^{k+1} - x^*\|^2$$

$$- 2\langle \nabla r(x_g^k) - y^{k+1}, x^{k+1} - x^*\rangle.$$

From optimality condition (18) it follows that $\nabla r(x^*) = y^*$ and hence

$$\frac{1}{\eta}\|x^{k+1} - x^*\|^2 \le \frac{1}{\eta}\|x^k - x^*\|^2 + \alpha\|x_g^k - x^*\|^2 - \alpha\|x^{k+1} - x^*\|^2 - \frac{1}{\eta}\|x^{k+1} - x^k\|^2$$

$$- 2\langle \nabla r(x_g^k) - \nabla r(x^*), x^{k+1} - x^*\rangle + 2\langle y^{k+1} - y^*, x^{k+1} - x^*\rangle.$$

Using lemma 5 we get

$$\frac{1}{\eta}\|x^{k+1} - x^*\|^2 \le \frac{1}{\eta}\|x^k - x^*\|^2 + \alpha\|x_g^k - x^*\|^2 - \alpha\|x^{k+1} - x^*\|^2 - \frac{1}{2\eta}\|x^{k+1} - x^k\|^2$$

$$- \frac{\eta\delta}{4}\|y^{k+1} - y^*\|^2 + \frac{\alpha}{4}\|x^{k+1} - x^*\|^2 + \mathrm{D}_r(x_g^k, x^*)$$

$$- 2\langle \nabla r(x_g^k) - \nabla r(x^*), x^{k+1} - x^*\rangle + 2\langle y^{k+1} - y^*, x^{k+1} - x^*\rangle$$

$$\le \frac{1}{\eta}\|x^k - x^*\|^2 + \alpha\|x_g^k - x^*\|^2 - \frac{3\alpha}{4}\|x^{k+1} - x^*\|^2 - \frac{1}{2\eta}\|x^{k+1} - x^k\|^2$$

$$- 2\langle \nabla r(x_g^k) - \nabla r(x^*), x^{k+1} - x^*\rangle + 2\langle y^{k+1} - y^*, x^{k+1} - x^*\rangle$$

$$+ \mathrm{D}_r(x_g^k, x^*) - \frac{\eta\delta}{4}\|y^{k+1} - y^*\|^2.$$

Using lines 6 and 12 of Algorithm 3 we get

$$\frac{1}{\eta}\|x^{k+1} - x^*\|^2 \le \frac{1}{\eta}\|x^k - x^*\|^2 + \alpha\|x_g^k - x^*\|^2 - \frac{3\alpha}{4}\|x^{k+1} - x^*\|^2$$

$$+ \frac{2(1-\tau)}{\tau}\langle \nabla r(x_g^k) - \nabla r(x^*), x_f^k - x_g^k\rangle - 2\langle \nabla r(x_g^k) - \nabla r(x^*), x_g^k - x^*\rangle$$

$$- \frac{2-\tau}{\tau}\langle \nabla r(x_g^k) - \nabla r(x^*), x_f^{k+1} - x_g^k\rangle - \frac{(2-\tau)^2}{8\eta\tau^2}\|x_f^{k+1} - x_g^k\|^2$$

$$+ \mathrm{D}_r(x_g^k, x^*) - \frac{\eta\delta}{4}\|y^{k+1} - y^*\|^2 + 2\langle y^{k+1} - y^*, x^{k+1} - x^*\rangle$$

$$\le \frac{1}{\eta}\|x^k - x^*\|^2 + \alpha\|x_g^k - x^*\|^2 - \frac{3\alpha}{4}\|x^{k+1} - x^*\|^2$$

$$+ \frac{2(1-\tau)}{\tau} \langle \nabla r(x_g^k) - \nabla r(x^*), x_f^k - x_g^k \rangle - 2\langle \nabla r(x_g^k) - \nabla r(x^*), x_g^k - x^* \rangle$$

$$- \frac{2-\tau}{\tau} \left( \langle \nabla r(x_g^k) - \nabla r(x^*), x_f^{k+1} - x_g^k \rangle + \frac{1}{8\eta\tau} \|x_f^{k+1} - x_g^k\|^2 \right)$$

$$+ D_r(x_g^k, x^*) - \frac{\eta\delta}{4} \|y^{k+1} - y^*\|^2 + 2\langle y^{k+1} - y^*, x^{k+1} - x^* \rangle.$$

Using $\frac{\mu}{2}$-strong convexity and $L$-smoothness of $r(x)$ and $\eta$ defined by (23) we get

$$\frac{1}{\eta}\|x^{k+1} - x^*\|^2 \le \frac{1}{\eta}\|x^k - x^*\|^2 + \alpha\|x_g^k - x^*\|^2 - \frac{3\alpha}{4}\|x^{k+1} - x^*\|^2$$

$$+ \frac{2(1-\tau)}{\tau}(D_r(x_f^k, x^*) - D_r(x_g^k, x^*)) - 2D_r(x_g^k, x^*) - \frac{\mu}{2}\|x_g^k - x^*\|^2$$

$$- \frac{2-\tau}{\tau} \left( \langle \nabla r(x_g^k) - \nabla r(x^*), x_f^{k+1} - x_g^k \rangle + \frac{L}{2}\|x_f^{k+1} - x_g^k\|^2 \right)$$

$$+ D_r(x_g^k, x^*) - \frac{\eta\delta}{4}\|y^{k+1} - y^*\|^2 + 2\langle y^{k+1} - y^*, x^{k+1} - x^* \rangle$$

$$\le \frac{1}{\eta}\|x^k - x^*\|^2 + \left(\alpha - \frac{\mu}{2}\right)\|x_g^k - x^*\|^2 - \frac{3\alpha}{4}\|x^{k+1} - x^*\|^2$$

$$+ \frac{2(1-\tau)}{\tau}(D_r(x_f^k, x^*) - D_r(x_g^k, x^*)) - \frac{2-\tau}{\tau}(D_r(x_f^{k+1}, x^*) - D_r(x_g^k, x^*))$$

$$- D_r(x_g^k, x^*) - \frac{\eta\delta}{4}\|y^{k+1} - y^*\|^2 + 2\langle y^{k+1} - y^*, x^{k+1} - x^* \rangle$$

$$= \frac{1}{\eta}\|x^k - x^*\|^2 + \left(\alpha - \frac{\mu}{2}\right)\|x_g^k - x^*\|^2 - \frac{3\alpha}{4}\|x^{k+1} - x^*\|^2$$

$$+ \frac{2(1-\tau)}{\tau}D_r(x_f^k, x^*) - \frac{2-\tau}{\tau}D_r(x_f^{k+1}, x^*)$$

$$- \frac{\eta\delta}{4}\|y^{k+1} - y^*\|^2 + 2\langle y^{k+1} - y^*, x^{k+1} - x^* \rangle.$$

Using $\alpha$ defined by (21) we get

$$\frac{1}{\eta}\|x^{k+1} - x^*\|^2 \le \frac{1}{\eta}\|x^k - x^*\|^2 - \frac{3\alpha}{4}\|x^{k+1} - x^*\|^2 + \frac{2(1-\tau)}{\tau}D_r(x_f^k, x^*) - \frac{2-\tau}{\tau}D_r(x_f^{k+1}, x^*)$$

$$- \frac{\eta\delta}{4}\|y^{k+1} - y^*\|^2 + 2\langle y^{k+1} - y^*, x^{k+1} - x^* \rangle.$$

$\square$

**Lemma 7.** *For all $y_1, y_2 \in \mathbb{R}^{nd}$ and $z_1, z_2 \in \text{range}\mathbf{W}$ the following inequality holds:*

$$D_h((y_1, z_1), (y_2, z_2)) \le \left(\frac{2}{\mu} + \frac{\nu}{2}\right)\|y_1 - y_2\|^2 + \frac{2}{\mu}\|z_1 - z_2\|^2.$$

*Proof.* It follows from from the definition of $D_h$:

$$D_h((y_1, z_1), (y_2, z_2)) = \frac{1}{\mu}\|y_1 + z_1 - y_2 - z_2\|^2 + \frac{\nu}{2}\|y_1 - y_2\|^2$$

$$\le \left(\frac{2}{\mu} + \frac{\nu}{2}\right)\|y_1 - y_2\|^2 + \frac{2}{\mu}\|z_1 - z_2\|^2.$$

$\square$

**Lemma 8.** *Let $\theta$ satisfy*

$$\theta \le \left[\sigma\left(\frac{4}{\mu} + \nu\right)\right]^{-1}. \tag{25}$$

*Let $\lambda$ satisfy*

$$\lambda \le \left[\frac{4\sigma\lambda_{\max}(\mathbf{W})}{\mu}\right]^{-1}. \tag{26}$$

*Let $\beta$ satisfy*

$$\beta \leq \min\left\{\frac{1}{\mu}, \frac{\nu}{3}\right\}. \tag{27}$$

*Let $\gamma$ satisfy*

$$\gamma \leq \lambda_{\min}^+(\mathbf{W})\beta. \tag{28}$$

*Then the following inequality holds:*

$$\left\|\begin{bmatrix} y^{k+1} - y^* \\ z^{k+1} - z^* \end{bmatrix}\right\|_{\mathbf{M}}^2 \leq \left\|\begin{bmatrix} y^k - y^* \\ z^k - z^* \end{bmatrix}\right\|_{\mathbf{M}}^2 - (\beta - 2\nu)\|y^{k+1} - y^*\|^2 - \gamma\|z^{k+1} - z^*\|_{\mathbf{W}^\dagger}^2 \tag{29}$$
$$+ \frac{2(1-\sigma)}{\sigma}\mathrm{D}_h((y_f^k, z_f^k), (y^*, z^*)) - \frac{2}{\sigma}\mathrm{D}_h((y_f^{k+1}, z_f^{k+1}), (y^*, z^*))$$
$$- 2\langle x^{k+1} - x^*, y^{k+1} - y^*\rangle,$$

*where $\mathbf{M} \in \mathbb{R}^{2nd \times 2nd}$ is a matrix defined by*

$$\mathbf{M} = \begin{bmatrix} \frac{1}{\theta}\mathbf{I} & 0 \\ 0 & \frac{1}{\lambda}\mathbf{W}^\dagger \end{bmatrix}. \tag{30}$$

*Proof.* Using line 10 of Algorithm 3 we get

$$\frac{1}{\theta}\|y^{k+1} - y^*\|^2 = \frac{1}{\theta}\|y^k - y^*\|^2 + \frac{2}{\theta}\langle y^{k+1} - y^k, y^{k+1} - y^*\rangle - \frac{1}{\theta}\|y^{k+1} - y^k\|^2$$
$$= \frac{1}{\theta}\|y^k - y^*\|^2 - \frac{1}{\theta}\|y^{k+1} - y^k\|^2 + 2\beta\langle y_g^k - y^{k+1}, y^{k+1} - y^*\rangle$$
$$- 2\langle \nabla_y h(y_g^k, z_g^k) - \nu y^{k+1} + x^{k+1}, y^{k+1} - y^*\rangle$$
$$= \frac{1}{\theta}\|y^k - y^*\|^2 - \frac{1}{\theta}\|y^{k+1} - y^k\|^2 + 2\beta\langle y_g^k - y^*, y^{k+1} - y^*\rangle - 2\beta\|y^{k+1} - y^*\|^2$$
$$- 2\langle \nabla_y h(y_g^k, z_g^k) - \nu y^{k+1} + x^{k+1}, y^{k+1} - y^*\rangle$$
$$\leq \frac{1}{\theta}\|y^k - y^*\|^2 + \beta\|y_g^k - y^*\|^2 - \beta\|y^{k+1} - y^*\|^2 - \frac{1}{\theta}\|y^{k+1} - y^k\|^2$$
$$- 2\langle \nabla_y h(y_g^k, z_g^k) - \nu y^{k+1} + x^{k+1}, y^{k+1} - y^*\rangle.$$

From optimality condition (19) it follows that $x^* = -\frac{2}{\mu}(y^* + z^*) = -\nabla_y h(y^*, z^*) + \nu y^*$ and hence

$$\frac{1}{\theta}\|y^{k+1} - y^*\|^2 \leq \frac{1}{\theta}\|y^k - y^*\|^2 + \beta\|y_g^k - y^*\|^2 - \beta\|y^{k+1} - y^*\|^2 - \frac{1}{\theta}\|y^{k+1} - y^k\|^2$$
$$- 2\langle \nabla_y h(y_g^k, z_g^k) - \nabla_y h(y^*, z^*), y^{k+1} - y^*\rangle$$
$$+ 2\nu\|y^{k+1} - y^*\|^2 - 2\langle x^{k+1} - x^*, y^{k+1} - y^*\rangle$$
$$= \frac{1}{\theta}\|y^k - y^*\|^2 - (\beta - 2\nu)\|y^{k+1} - y^*\|^2 + \beta\|y_g^k - y^*\|^2 - \frac{1}{\theta}\|y^{k+1} - y^k\|^2$$
$$- 2\langle \nabla_y h(y_g^k, z_g^k) - \nabla_y h(y^*, z^*), y^{k+1} - y^*\rangle - 2\langle x^{k+1} - x^*, y^{k+1} - y^*\rangle.$$

Using lines 7 and 13 of Algorithm 3 we get

$$\frac{1}{\theta}\|y^{k+1} - y^*\|^2 \leq \frac{1}{\theta}\|y^k - y^*\|^2 - (\beta - 2\nu)\|y^{k+1} - y^*\|^2 + \beta\|y_g^k - y^*\|^2$$
$$+ \frac{2(1-\sigma)}{\sigma}\langle \nabla_y h(y_g^k, z_g^k) - \nabla_y h(y^*, z^*), y_f^k - y_g^k\rangle$$
$$- \frac{2}{\sigma}\langle \nabla_y h(y_g^k, z_g^k) - \nabla_y h(y^*, z^*), y_f^{k+1} - y_g^k\rangle - \frac{1}{\theta\sigma^2}\|y_f^{k+1} - y_g^k\|^2$$
$$- 2\langle \nabla_y h(y_g^k, z_g^k) - \nabla_y h(y^*, z^*), y_g^k - y^*\rangle - 2\langle x^{k+1} - x^*, y^{k+1} - y^*\rangle.$$

Using $\theta$ defined by (25) we get

$$\frac{1}{\theta}\|y^{k+1} - y^*\|^2 \leq \frac{1}{\theta}\|y^k - y^*\|^2 - (\beta - 2\nu)\|y^{k+1} - y^*\|^2 + \beta\|y_g^k - y^*\|^2 \tag{31}$$

$$+ \frac{2(1-\sigma)}{\sigma}\langle \nabla_y h(y_g^k, z_g^k) - \nabla_y h(y^*, z^*), y_f^k - y_g^k\rangle$$

$$- \frac{2}{\sigma}\left(\langle \nabla_y h(y_g^k, z_g^k) - \nabla_y h(y^*, z^*), y_f^{k+1} - y_g^k\rangle + \left(\frac{2}{\mu} + \frac{\nu}{2}\right)\|y_f^{k+1} - y_g^k\|^2\right)$$

$$- 2\langle \nabla_y h(y_g^k, z_g^k) - \nabla_y h(y^*, z^*), y_g^k - y^*\rangle - 2\langle x^{k+1} - x^*, y^{k+1} - y^*\rangle.$$

Using line 11 of Algorithm 3 we get

$$\frac{1}{\lambda}\|z^{k+1} - z^*\|_{\mathbf{W}^\dagger}^2 = \frac{1}{\lambda}\|z^k - z^*\|_{\mathbf{W}^\dagger}^2 + \frac{2}{\lambda}\langle z^{k+1} - z^k, \mathbf{W}^\dagger(z^{k+1} - z^*)\rangle - \frac{1}{\lambda}\|z^{k+1} - z^k\|_{\mathbf{W}^\dagger}^2$$

$$= \frac{1}{\lambda}\|z^k - z^*\|_{\mathbf{W}^\dagger}^2 - \frac{1}{\lambda}\|z^{k+1} - z^k\|_{\mathbf{W}^\dagger}^2 + 2\gamma\langle z_g^k - z^{k+1}, \mathbf{W}^\dagger(z^{k+1} - z^*)\rangle$$

$$- 2\langle \mathbf{W}\nabla_z h(y_g^k, z_g^k), \mathbf{W}^\dagger(z^{k+1} - z^*)\rangle$$

$$= \frac{1}{\lambda}\|z^k - z^*\|_{\mathbf{W}^\dagger}^2 + 2\gamma\langle z_g^k - z^*, \mathbf{W}^\dagger(z^{k+1} - z^*)\rangle - 2\gamma\|z^{k+1} - z^*\|_{\mathbf{W}^\dagger}^2$$

$$- \frac{1}{\lambda}\|z^{k+1} - z^k\|_{\mathbf{W}^\dagger}^2 - 2\langle \mathbf{W}\nabla_z h(y_g^k, z_g^k), \mathbf{W}^\dagger(z^{k+1} - z^*)\rangle$$

$$\leq \frac{1}{\lambda}\|z^k - z^*\|_{\mathbf{W}^\dagger}^2 + \gamma\|z_g^k - z^*\|_{\mathbf{W}^\dagger}^2 - \gamma\|z^{k+1} - z^*\|_{\mathbf{W}^\dagger}^2 - \frac{1}{\lambda}\|z^{k+1} - z^k\|_{\mathbf{W}^\dagger}^2$$

$$- 2\langle \mathbf{W}\nabla_z h(y_g^k, z_g^k), \mathbf{W}^\dagger(z^{k+1} - z^*)\rangle.$$

From optimality condition (20) it follows that $\mathbf{W}\nabla_z h(y^*, z^*) = 0$ and hence

$$\frac{1}{\lambda}\|z^{k+1} - z^*\|_{\mathbf{W}^\dagger}^2 \leq \frac{1}{\lambda}\|z^k - z^*\|_{\mathbf{W}^\dagger}^2 + \gamma\|z_g^k - z^*\|_{\mathbf{W}^\dagger}^2 - \gamma\|z^{k+1} - z^*\|_{\mathbf{W}^\dagger}^2 - \frac{1}{\lambda}\|z^{k+1} - z^k\|_{\mathbf{W}^\dagger}^2$$

$$- 2\langle \mathbf{W}(\nabla_z h(y_g^k, z_g^k) - \nabla_z h(y^*, z^*)), \mathbf{W}^\dagger(z^{k+1} - z^*)\rangle$$

$$= \frac{1}{\lambda}\|z^k - z^*\|_{\mathbf{W}^\dagger}^2 + \gamma\|z_g^k - z^*\|_{\mathbf{W}^\dagger}^2 - \gamma\|z^{k+1} - z^*\|_{\mathbf{W}^\dagger}^2 - \frac{1}{\lambda}\|z^{k+1} - z^k\|_{\mathbf{W}^\dagger}^2$$

$$- 2\langle \nabla_z h(y_g^k, z_g^k) - \nabla_z h(y^*, z^*), \mathbf{W}\mathbf{W}^\dagger(z^{k+1} - z^*)\rangle.$$

It's easy to observe that $z^k, z^* \in \mathrm{range}\mathbf{W}$ for all $k = 0, 1, 2, \ldots$, which implies

$$\mathbf{W}\mathbf{W}^\dagger(z^{k+1} - z^*) = z^{k+1} - z^* \text{ and } \|z^{k+1} - z^k\|_{\mathbf{W}^\dagger}^2 \geq \frac{1}{\lambda_{\max}(\mathbf{W})}\|z^{k+1} - z^k\|^2.$$

Hence,

$$\frac{1}{\lambda}\|z^{k+1} - z^*\|_{\mathbf{W}^\dagger}^2 \leq \frac{1}{\lambda}\|z^k - z^*\|_{\mathbf{W}^\dagger}^2 - \gamma\|z^{k+1} - z^*\|_{\mathbf{W}^\dagger}^2 + \frac{\gamma}{\lambda_{\min}^+(\mathbf{W})}\|z_g^k - z^*\|^2$$

$$- \frac{1}{\lambda \cdot \lambda_{\max}(\mathbf{W})}\|z^{k+1} - z^k\|^2 - 2\langle \nabla_z h(y_g^k, z_g^k) - \nabla_z h(y^*, z^*), z^{k+1} - z^*\rangle.$$

Using lines 8 and 14 of Algorithm 3 we get

$$\frac{1}{\lambda}\|z^{k+1} - z^*\|_{\mathbf{W}^\dagger}^2 \leq \frac{1}{\lambda}\|z^k - z^*\|_{\mathbf{W}^\dagger}^2 - \gamma\|z^{k+1} - z^*\|_{\mathbf{W}^\dagger}^2 + \frac{\gamma}{\lambda_{\min}^+(\mathbf{W})}\|z_g^k - z^*\|_{\mathbf{W}^\dagger}^2$$

$$+ \frac{2(1-\sigma)}{\sigma}\langle \nabla_z h(y_g^k, z_g^k) - \nabla_z h(y^*, z^*), z_f^k - z_g^k\rangle$$

$$- \frac{2}{\sigma}\langle \nabla_z h(y_g^k, z_g^k) - \nabla_z h(y^*, z^*), z_f^{k+1} - z_g^k\rangle - \frac{1}{\lambda\sigma^2\lambda_{\max}(\mathbf{W})}\|z_f^{k+1} - z_g^k\|^2$$

$$- 2\langle \nabla_z h(y_g^k, z_g^k) - \nabla_z h(y^*, z^*), z_g^k - z^*\rangle.$$

Using $\lambda$ defined by (26) we get

$$\frac{1}{\lambda}\|z^{k+1}-z^*\|^2_{\mathbf{W}^\dagger} \le \frac{1}{\lambda}\|z^k-z^*\|^2_{\mathbf{W}^\dagger} - \gamma\|z^{k+1}-z^*\|^2_{\mathbf{W}^\dagger} + \frac{\gamma}{\lambda^+_{\min}(\mathbf{W})}\|z^k_g-z^*\|^2_{\mathbf{W}^\dagger} \qquad (32)$$

$$+ \frac{2(1-\sigma)}{\sigma}\langle \nabla_z h(y^k_g,z^k_g) - \nabla_z h(y^*,z^*), z^k_f - z^k_g \rangle$$

$$- \frac{2}{\sigma}\left( \langle \nabla_z h(y^k_g,z^k_g) - \nabla_z h(y^*,z^*), z^{k+1}_f - z^k_g \rangle - \frac{2}{\mu}\|z^{k+1}_f - z^k_g\|^2 \right)$$

$$- 2\langle \nabla_z h(y^k_g,z^k_g) - \nabla_z h(y^*,z^*), z^k_g - z^* \rangle.$$

After combining (31) and (32) we get

$$\left\| \begin{bmatrix} y^{k+1}-y^* \\ z^{k+1}-z^* \end{bmatrix} \right\|^2_{\mathbf{M}} \le \left\| \begin{bmatrix} y^k-y^* \\ z^k-z^* \end{bmatrix} \right\|^2_{\mathbf{M}} - (\beta-2\nu)\|y^{k+1}-y^*\|^2 - \gamma\|z^{k+1}-z^*\|^2_{\mathbf{W}^\dagger}$$

$$+ \beta\|y^k_g-y^*\|^2 + \frac{\gamma}{\lambda^+_{\min}(\mathbf{W})}\|z^k_g-z^*\|^2$$

$$+ \frac{2(1-\sigma)}{\sigma}\left\langle \nabla h(y^k_g,z^k_g) - \nabla h(y^*,z^*), \begin{bmatrix} y^k_f \\ z^k_f \end{bmatrix} - \begin{bmatrix} y^k_g \\ z^k_g \end{bmatrix} \right\rangle$$

$$- \frac{2}{\sigma}\left\langle \nabla h(y^k_g,z^k_g) - \nabla h(y^*,z^*), \begin{bmatrix} y^{k+1}_f \\ z^{k+1}_f \end{bmatrix} - \begin{bmatrix} y^k_g \\ z^k_g \end{bmatrix} \right\rangle$$

$$- \frac{2}{\sigma}\left( \left(\frac{2}{\mu}+\frac{\nu}{2}\right)\|y^{k+1}_f - z^k_g\|^2 + \frac{2}{\mu}\|z^{k+1}_f - z^k_g\|^2 \right)$$

$$- 2\left\langle \nabla h(y^k_g,z^k_g) - \nabla h(y^*,z^*), \begin{bmatrix} y^k_g \\ z^k_g \end{bmatrix} - \begin{bmatrix} y^* \\ z^* \end{bmatrix} \right\rangle - 2\langle x^{k+1}-x^*, y^{k+1}-y^* \rangle,$$

where $\mathbf{M} \in \mathbb{R}^{2nd\times 2nd}$ is a matrix defined by (30). Using convexity of $h(y,z)$ and the fact that
$\nabla h(y,z) = \begin{bmatrix} \frac{2}{\mu}(y+z) + \nu y \\ \frac{2}{\mu}(y+z) \end{bmatrix}$ we get

$$\left\| \begin{bmatrix} y^{k+1}-y^* \\ z^{k+1}-z^* \end{bmatrix} \right\|^2_{\mathbf{M}} \le \left\| \begin{bmatrix} y^k-y^* \\ z^k-z^* \end{bmatrix} \right\|^2_{\mathbf{M}} - (\beta-2\nu)\|y^{k+1}-y^*\|^2 - \gamma\|z^{k+1}-z^*\|^2_{\mathbf{W}^\dagger}$$

$$+ \beta\|y^k_g-y^*\|^2 + \frac{\gamma}{\lambda^+_{\min}(\mathbf{W})}\|z^k_g-z^*\|^2$$

$$+ \frac{2(1-\sigma)}{\sigma}\left[ D_h((y^k_f,z^k_f),(y^*,z^*)) - D_h((y^k_g,z^k_g),(y^*,z^*)) \right]$$

$$- \frac{2}{\sigma}\left\langle \nabla h(y^k_g,z^k_g) - \nabla h(y^*,z^*), \begin{bmatrix} y^{k+1}_f \\ z^{k+1}_f \end{bmatrix} - \begin{bmatrix} y^k_g \\ z^k_g \end{bmatrix} \right\rangle$$

$$- \frac{2}{\sigma}\left( \left(\frac{2}{\mu}+\frac{\nu}{2}\right)\|y^{k+1}_f - z^k_g\|^2 + \frac{2}{\mu}\|z^{k+1}_f - z^k_g\|^2 \right)$$

$$- \frac{4}{\mu}\|y^k_g + z^k_g - y^* - z^*\|^2 - 2\nu\|y^k_g - y^*\|^2 - 2\langle x^{k+1}-x^*, y^{k+1}-y^* \rangle.$$

Using lemma 7 we can obtain

$$\left\|\begin{bmatrix} y^{k+1}-y^* \\ z^{k+1}-z^* \end{bmatrix}\right\|^2_{\mathbf{M}} \le \left\|\begin{bmatrix} y^k-y^* \\ z^k-z^* \end{bmatrix}\right\|^2_{\mathbf{M}} - (\beta-2\nu)\|y^{k+1}-y^*\|^2 - \gamma\|z^{k+1}-z^*\|^2_{\mathbf{W}^\dagger}$$
$$+ (\beta-2\nu)\|y_g^k-y^*\|^2 + \frac{\gamma}{\lambda^+_{\min}(\mathbf{W})}\|z_g^k-z^*\|^2 - \frac{4}{\mu}\|y_g^k+z_g^k-y^*-z^*\|^2$$
$$+ \frac{2(1-\sigma)}{\sigma}\left[\mathrm{D}_h((y_f^k,z_f^k),(y^*,z^*)) - \mathrm{D}_h((y_g^k,z_g^k),(y^*,z^*))\right]$$
$$- \frac{2}{\sigma}\left[\mathrm{D}_h((y_f^{k+1},z_f^{k+1}),(y^*,z^*)) - \mathrm{D}_h((y_g^k,z_g^k),(y^*,z^*))\right]$$
$$- 2\langle x^{k+1}-x^*, y^{k+1}-y^*\rangle$$
$$= \left\|\begin{bmatrix} y^k-y^* \\ z^k-z^* \end{bmatrix}\right\|^2_{\mathbf{M}} - (\beta-2\nu)\|y^{k+1}-y^*\|^2 - \gamma\|z^{k+1}-z^*\|^2_{\mathbf{W}^\dagger}$$
$$+ (\beta-2\nu)\|y_g^k-y^*\|^2 + \frac{\gamma}{\lambda^+_{\min}(\mathbf{W})}\|z_g^k-z^*\|^2 - \frac{4}{\mu}\|y_g^k+z_g^k-y^*-z^*\|^2$$
$$+ \frac{2(1-\sigma)}{\sigma}\mathrm{D}_h((y_f^k,z_f^k),(y^*,z^*)) - \frac{2}{\sigma}\mathrm{D}_h((y_f^{k+1},z_f^{k+1}),(y^*,z^*))$$
$$+ 2\mathrm{D}_h((y_g^k,z_g^k),(y^*,z^*)) - 2\langle x^{k+1}-x^*, y^{k+1}-y^*\rangle$$
$$= \left\|\begin{bmatrix} y^k-y^* \\ z^k-z^* \end{bmatrix}\right\|^2_{\mathbf{M}} - (\beta-2\nu)\|y^{k+1}-y^*\|^2 - \gamma\|z^{k+1}-z^*\|^2_{\mathbf{W}^\dagger}$$
$$+ (\beta-2\nu)\|y_g^k-y^*\|^2 + \frac{\gamma}{\lambda^+_{\min}(\mathbf{W})}\|z_g^k-z^*\|^2 - \frac{4}{\mu}\|y_g^k+z_g^k-y^*-z^*\|^2$$
$$+ \frac{2(1-\sigma)}{\sigma}\mathrm{D}_h((y_f^k,z_f^k),(y^*,z^*)) - \frac{2}{\sigma}\mathrm{D}_h((y_f^{k+1},z_f^{k+1}),(y^*,z^*))$$
$$+ \frac{2}{\mu}\|y_g^k+z_g^k-y^*-z^*\|^2 + \nu\|y_g^k-y^*\|^2 - 2\langle x^{k+1}-x^*, y^{k+1}-y^*\rangle$$
$$= \left\|\begin{bmatrix} y^k-y^* \\ z^k-z^* \end{bmatrix}\right\|^2_{\mathbf{M}} - (\beta-2\nu)\|y^{k+1}-y^*\|^2 - \gamma\|z^{k+1}-z^*\|^2_{\mathbf{W}^\dagger}$$
$$+ \frac{2(1-\sigma)}{\sigma}\mathrm{D}_h((y_f^k,z_f^k),(y^*,z^*)) - \frac{2}{\sigma}\mathrm{D}_h((y_f^{k+1},z_f^{k+1}),(y^*,z^*))$$
$$+ (\beta-\nu)\|y_g^k-y^*\|^2 + \frac{\gamma}{\lambda^+_{\min}(\mathbf{W})}\|z_g^k-z^*\|^2 - \frac{2}{\mu}\|y_g^k+z_g^k-y^*-z^*\|^2$$
$$- 2\langle x^{k+1}-x^*, y^{k+1}-y^*\rangle.$$

Using $\gamma$ defined by (28) and the fact that $\beta \le \frac{1}{\mu}$ which follows from (27) we get

$$\left\|\begin{bmatrix} y^{k+1}-y^* \\ z^{k+1}-z^* \end{bmatrix}\right\|^2_{\mathbf{M}} \le \left\|\begin{bmatrix} y^k-y^* \\ z^k-z^* \end{bmatrix}\right\|^2_{\mathbf{M}} - (\beta-2\nu)\|y^{k+1}-y^*\|^2 - \gamma\|z^{k+1}-z^*\|^2_{\mathbf{W}^\dagger}$$
$$+ \frac{2(1-\sigma)}{\sigma}\mathrm{D}_h((y_f^k,z_f^k),(y^*,z^*)) - \frac{2}{\sigma}\mathrm{D}_h((y_f^{k+1},z_f^{k+1}),(y^*,z^*))$$
$$+ (\beta-\nu)\|y_g^k-y^*\|^2 + \beta\|z_g^k-z^*\|^2 - 2\beta\|y_g^k+z_g^k-y^*-z^*\|^2$$
$$- 2\langle x^{k+1}-x^*, y^{k+1}-y^*\rangle$$

$$\leq \left\| \begin{bmatrix} y^k - y^* \\ z^k - z^* \end{bmatrix} \right\|_{\mathbf{M}}^2 - (\beta - 2\nu)\|y^{k+1} - y^*\|^2 - \gamma\|z^{k+1} - z^*\|_{\mathbf{W}^\dagger}^2$$

$$+ \frac{2(1-\sigma)}{\sigma}\mathrm{D}_h((y_f^k, z_f^k), (y^*, z^*)) - \frac{2}{\sigma}\mathrm{D}_h((y_f^{k+1}, z_f^{k+1}), (y^*, z^*))$$

$$+ (\beta - \nu)\|y_g^k - y^*\|^2 + \beta\|z_g^k - z^*\|^2 - \beta\|z_g^k - z^*\|^2 + 2\beta\|y_g^k - y^*\|^2$$

$$- 2\langle x^{k+1} - x^*, y^{k+1} - y^* \rangle$$

$$= \left\| \begin{bmatrix} y^k - y^* \\ z^k - z^* \end{bmatrix} \right\|_{\mathbf{M}}^2 - (\beta - 2\nu)\|y^{k+1} - y^*\|^2 - \gamma\|z^{k+1} - z^*\|_{\mathbf{W}^\dagger}^2$$

$$+ \frac{2(1-\sigma)}{\sigma}\mathrm{D}_h((y_f^k, z_f^k), (y^*, z^*)) - \frac{2}{\sigma}\mathrm{D}_h((y_f^{k+1}, z_f^{k+1}), (y^*, z^*))$$

$$+ (3\beta - \nu)\|y_g^k - y^*\|^2 - 2\langle x^{k+1} - x^*, y^{k+1} - y^* \rangle.$$

Using the fact that $\beta \leq \frac{\nu}{3}$ which follows from (27) we get

$$\left\| \begin{bmatrix} y^{k+1} - y^* \\ z^{k+1} - z^* \end{bmatrix} \right\|_{\mathbf{M}}^2 \leq \left\| \begin{bmatrix} y^k - y^* \\ z^k - z^* \end{bmatrix} \right\|_{\mathbf{M}}^2 - (\beta - 2\nu)\|y^{k+1} - y^*\|^2 - \gamma\|z^{k+1} - z^*\|_{\mathbf{W}^\dagger}^2$$

$$+ \frac{2(1-\sigma)}{\sigma}\mathrm{D}_h((y_f^k, z_f^k), (y^*, z^*)) - \frac{2}{\sigma}\mathrm{D}_h((y_f^{k+1}, z_f^{k+1}), (y^*, z^*))$$

$$- 2\langle x^{k+1} - x^*, y^{k+1} - y^* \rangle.$$

$\square$

**Theorem 4.** *Let $\tau$ be defined by*

$$\tau = \frac{1}{2}\sqrt{\frac{\mu}{L}}.$$

*Let $\alpha$ be defined by*

$$\alpha = \frac{\mu}{2}.$$

*Let $\eta$ be defined by*

$$\eta = \frac{1}{2\sqrt{\mu L}}.$$

*Let $\sigma$ be defined by*

$$\sigma = \frac{1}{18}\sqrt{\frac{\mu\lambda_{\min}^+(\mathbf{W})}{L\lambda_{\max}(\mathbf{W})}}.$$

*Let $\nu$ be defined by*

$$\nu = \frac{3}{80L}.$$

*Let $\beta$ be defined by*

$$\beta = \frac{1}{80L}.$$

*Let $\theta$ be defined by*

$$\theta = \frac{18\sqrt{\mu L\lambda_{\max}(\mathbf{W})}}{5\sqrt{\lambda_{\min}^+(\mathbf{W})}}$$

*Let $\gamma$ be defined by*

$$\gamma = \frac{\lambda_{\min}^+(\mathbf{W})}{80L}.$$

*Let $\lambda$ be defined by*

$$\lambda = \frac{9\sqrt{\mu L}}{2\sqrt{\lambda_{\min}^+(\mathbf{W})\lambda_{\max}(\mathbf{W})}}.$$

*where $\mathbf{P} \in \mathbb{R}^{3nd \times 3nd}$ is a matrix defined by*

$$\mathbf{P} = \begin{bmatrix} \frac{1}{\eta}\mathbf{I} & 0 & 0 \\ 0 & \frac{1}{\theta}\mathbf{I} & 0 \\ 0 & 0 & \frac{1}{\lambda}\mathbf{W}^\dagger \end{bmatrix}. \tag{33}$$

*Let $\rho$ be defined by*

$$\rho = \frac{1}{18}\sqrt{\frac{\mu\lambda_{\min}^+(\mathbf{W})}{L\lambda_{\max}(\mathbf{W})}}.$$

*Let $\Psi^k$ be the following Lyapunov function:*

$$\Psi^k = (1+\rho)\left\|\begin{bmatrix} x^k - x^* \\ y^k - y^* \\ z^k - z^* \end{bmatrix}\right\|_{\mathbf{P}}^2 + \frac{(2-\tau)}{\tau}\mathrm{D}_r(x_f^k, x^*) + \frac{2}{\sigma}\mathrm{D}_h((y_f^k, z_f^k), (y^*, z^*)). \tag{34}$$

*Then the following inequality holds:*

$$\Psi^{k+1} \le \left(1 - \frac{1}{1+\rho^{-1}}\right)\Psi^k.$$

*Proof.* One can observe that conditions of lemma 6 and lemma 8 are satisfied. Hence we can combine (24) and (29) and get

$$\left\|\begin{bmatrix} x^{k+1} - x^* \\ y^{k+1} - y^* \\ z^{k+1} - z^* \end{bmatrix}\right\|_{\mathbf{P}}^2 = \frac{1}{\eta}\|x^{k+1} - y^*\|^2 + \left\|\begin{bmatrix} y^{k+1} - y^* \\ z^{k+1} - z^* \end{bmatrix}\right\|_{\mathbf{M}}^2$$

$$\le \frac{1}{\eta}\|x^k - x^*\|^2 - \frac{3\alpha}{4}\|x^{k+1} - x^*\|^2 + \frac{2(1-\tau)}{\tau}\mathrm{D}_r(x_f^k, x^*)$$

$$- \frac{2-\tau}{\tau}\mathrm{D}_r(x_f^{k+1}, x^*) - \frac{\eta\delta}{4}\|y^{k+1} - y^*\|^2 + 2\langle y^{k+1} - y^*, x^{k+1} - x^*\rangle$$

$$+ \left\|\begin{bmatrix} y^k - y^* \\ z^k - z^* \end{bmatrix}\right\|_{\mathbf{M}}^2 - (\beta - 2\nu)\|y^{k+1} - y^*\|^2 - \gamma\|z^{k+1} - z^*\|_{\mathbf{W}^\dagger}^2$$

$$+ \frac{2(1-\sigma)}{\sigma}\mathrm{D}_h((y_f^k, z_f^k), (y^*, z^*)) - \frac{2}{\sigma}\mathrm{D}_h((y_f^{k+1}, z_f^{k+1}), (y^*, z^*))$$

$$- 2\langle x^{k+1} - x^*, y^{k+1} - y^*\rangle$$

$$= \left\|\begin{bmatrix} x^k - x^* \\ y^k - y^* \\ z^k - z^* \end{bmatrix}\right\|_{\mathbf{P}}^2 - \frac{3\alpha}{4}\|x^{k+1} - x^*\|^2 - \left(\frac{\eta\delta}{4} + \beta - 2\nu\right)\|y^{k+1} - y^*\|^2$$

$$-\gamma\|z^{k+1}-z^*\|^2_{\mathbf{W}^\dagger}+\frac{2(1-\tau)}{\tau}D_r(x_f^k,x^*)-\frac{2-\tau}{\tau}D_r(x_f^{k+1},x^*)$$
$$+\frac{2(1-\sigma)}{\sigma}D_h((y_f^k,z_f^k),(y^*,z^*))-\frac{2}{\sigma}D_h((y_f^{k+1},z_f^{k+1}),(y^*,z^*)),$$

where $\mathbf{P}\in\mathbb{R}^{3nd\times 3nd}$ is a matrix defined by (33). From (22) it follows that

$$\frac{\eta\delta}{4}=\min\left\{\frac{1}{8\sqrt{\mu L}},\frac{1}{8L}\right\}=\frac{1}{8L},$$

and hence, using choice of $\alpha$, $\beta$ and $\nu$, we get

$$\left\|\begin{bmatrix}x^{k+1}-x^*\\y^{k+1}-y^*\\z^{k+1}-z^*\end{bmatrix}\right\|^2_{\mathbf{P}}\leq\left\|\begin{bmatrix}x^k-x^*\\y^k-y^*\\z^k-z^*\end{bmatrix}\right\|^2_{\mathbf{P}}-\frac{3\mu}{8}\|x^{k+1}-x^*\|^2-\left(\frac{1}{8L}+\frac{1}{80L}-\frac{6}{80L}\right)\|y^{k+1}-y^*\|^2$$

$$-\frac{\lambda^+_{\min}(\mathbf{W})}{80L}\|z^{k+1}-z^*\|^2_{\mathbf{W}^\dagger}+\frac{2(1-\tau)}{\tau}D_r(x_f^k,x^*)-\frac{2-\tau}{\tau}D_r(x_f^{k+1},x^*)$$

$$+\frac{2(1-\sigma)}{\sigma}D_h((y_f^k,z_f^k),(y^*,z^*))-\frac{2}{\sigma}D_h((y_f^{k+1},z_f^{k+1}),(y^*,z^*))$$

$$\leq\left\|\begin{bmatrix}x^k-x^*\\y^k-y^*\\z^k-z^*\end{bmatrix}\right\|^2_{\mathbf{P}}-\min\left\{\frac{3\eta\mu}{8},\frac{\theta}{16L},\frac{\lambda\cdot\lambda^+_{\min}(\mathbf{W})}{80L}\right\}\left\|\begin{bmatrix}x^{k+1}-x^*\\y^{k+1}-y^*\\z^{k+1}-z^*\end{bmatrix}\right\|^2_{\mathbf{P}}$$

$$+\left(1-\frac{\tau}{2}\right)\frac{(2-\tau)}{\tau}D_r(x_f^k,x^*)-\frac{2-\tau}{\tau}D_r(x_f^{k+1},x^*)$$

$$+(1-\sigma)\frac{2}{\sigma}D_h((y_f^k,z_f^k),(y^*,z^*))-\frac{2}{\sigma}D_h((y_f^{k+1},z_f^{k+1}),(y^*,z^*))$$

$$=\left\|\begin{bmatrix}x^k-x^*\\y^k-y^*\\z^k-z^*\end{bmatrix}\right\|^2_{\mathbf{P}}-\min\left\{\frac{3}{16}\sqrt{\frac{\mu}{L}},\frac{9\sqrt{\mu\lambda_{\max}(\mathbf{W})}}{40\sqrt{L\lambda^+_{\min}(\mathbf{W})}},\frac{9\sqrt{\mu\lambda^+_{\min}(\mathbf{W})}}{160\sqrt{L\lambda_{\max}(\mathbf{W})}}\right\}\left\|\begin{bmatrix}x^{k+1}-x^*\\y^{k+1}-y^*\\z^{k+1}-z^*\end{bmatrix}\right\|^2_{\mathbf{P}}$$

$$+\left(1-\frac{1}{4}\sqrt{\frac{\mu}{L}}\right)\frac{(2-\tau)}{\tau}D_r(x_f^k,x^*)-\frac{2-\tau}{\tau}D_r(x_f^{k+1},x^*)$$

$$+\left(1-\frac{1}{18}\sqrt{\frac{\mu\lambda^+_{\min}(\mathbf{W})}{L\lambda_{\max}(\mathbf{W})}}\right)\frac{2}{\sigma}D_h((y_f^k,z_f^k),(y^*,z^*))-\frac{2}{\sigma}D_h((y_f^{k+1},z_f^{k+1}),(y^*,z^*))$$

$$\leq\left\|\begin{bmatrix}x^k-x^*\\y^k-y^*\\z^k-z^*\end{bmatrix}\right\|^2_{\mathbf{P}}-\rho\left\|\begin{bmatrix}x^{k+1}-x^*\\y^{k+1}-y^*\\z^{k+1}-z^*\end{bmatrix}\right\|^2_{\mathbf{P}}$$

$$+(1-\rho)\frac{(2-\tau)}{\tau}D_r(x_f^k,x^*)-\frac{2-\tau}{\tau}D_r(x_f^{k+1},x^*)$$

$$+(1-\rho)\frac{2}{\sigma}D_h((y_f^k,z_f^k),(y^*,z^*))-\frac{2}{\sigma}D_h((y_f^{k+1},z_f^{k+1}),(y^*,z^*)).$$

After rearranging and using definition of $\Psi^k$ (34) we get

$$\Psi^{k+1} \leq \left\| \begin{bmatrix} x^k - x^* \\ y^k - y^* \\ z^k - z^* \end{bmatrix} \right\|_{\mathbf{P}}^2 + (1-\rho)\frac{(2-\tau)}{\tau}D_r(x_f^k, x^*)$$

$$+ (1-\rho)\frac{2}{\sigma}D_h((y_f^k, z_f^k), (y^*, z^*))$$

$$\leq \left( 1 - \frac{1}{1+\rho^{-1}} \right)\Psi^k.$$

$\square$