[Reviews · NeurIPS 2020]

Review 1

Summary and Contributions: The paper proposes a new algorithm, APAPC, for decentralized gossip-based optimization of strongly convex objectives that is optimal both in gradient complexity and in communication-rounds complexity. That is, it both takes advantage of acceleration in terms of the condition number of the objective and depends optimally on the spectral gap of the gossip matrix used by the decentralized optimization method.

Strengths: The theory seems solid, and the problem that the paper sets out to address is well motivated by the desire to avoid dual gradients of the local loss functions. The proposed algorithm appears to compare well to the competitors.

Weaknesses: The proposed algorithm seems like a straightforward combination of the prior approaches that were optimal for gradient complexity and communication-round complexity via the method of operator splitting. So one could perhaps criticize this work as being incremental.

Correctness: The theoretical results appear to be correct. While the experiments are valid methodologically, what is missing here is an actual wall-clock-time comparison between the different algorithms for training on a real decentralized network. As it stands, we don't know how this approach would compare with MSDA and the other competitors in a practical setting, because it is not clear how the "# of communication rounds" and "# of gradients" computed translate to actual time taken by the decentralized solver. The networks chosen for evaluating the decentralized method seem a bit unnatural. Were these used in some previous work? If so, it should be cited; if not, why wasn't a network topology from a standard decentralized learning paper (such as a hypercube or cycle) tried?

Clarity: The paper is written well, but I worry that readers who are unfamiliar with the monotone operator framework will have difficulty understanding it. A lot of definitions are dropped on the reader at the top of page 5, without really giving much intuition about what these definitions are doing. But perhaps this is unavoidable.

Relation to Prior Work: The paper does a good job of situating itself in relation to prior work.

Reproducibility: Yes

Additional Feedback:


Review 2

Summary and Contributions: This paper considered the problem of decentralized optimization of smooth and strongly convex objectives, and proposed the first algorithm that is optimal in both communication complexity and gradient evaluation complexity, and does not require evaluation on dual gradients. Empirical comparison with state-of-the-art methods are also presented, demonstrating the effectiveness of the proposed approach.

Strengths: Clear theoretical justification and empirical demonstration of the effectiveness of the proposed approach.

Weaknesses: NA

Correctness: Theoretical results are clearly stated and seem correct. Empirical results look reasonable.

Clarity: This paper is well written.

Relation to Prior Work: Related work are extensively discussed.

Reproducibility: Yes

Additional Feedback:


Review 3

Summary and Contributions: The paper considers smooth and strongly-convex decentralized optimization problems. The authors propose a novel primal gossip-based optimization algorithm (OPAPC) which is provably optimal w.r.t. relevant oracle complexity lower bounds. Existing optimal algorithms for this setting rely on access to dual gradient which, in absence of any structure, can be very expensive. The suggested method uses primal (1st order) access exclusively---forming the main contribution of the paper. The main algorithmic idea is to use a generalized forward backward-like algorithm with the corresponding positive-definite operator being chosen according to the gossip matrix (so as to comply with the underlying decentralized structure). Acceleration w.r.t. the condition number of the individual functions and of the network graph is then obtained through Nesterov extrapolation and Chebyshev polynomials, respectively. --- update after the authors' response --- I have read the authors' response and other reviewers' comments carefully and taken them into account in my final evaluation. Good luck!

Strengths: The decentralized setting has gain a lot of popularity in ML in recent year for natural reasons. The suggested method OPAPC closes the theoretical gap for the class of primal algorithms for this setting and w.r.t. the oracle complexity bounds derived in Scaman et al. (2017). Although the actual optimality is obtained through integration of existing known algorithmic building blocks, the main ingredient (PAPC) is simple and elegant. Lastly, preliminary empirical results seem promising.

Weaknesses: As mentioned earlier, all the ingredients of the proposed algorithm are well-known and have been studied in previous works--in this sense, the technical contribution of the work is somewhat incremental. The choice of the definite-positive operator according to the gossip matrix is the valuable bit. My other two reservations are general to the context of the work: -- The only *physical* constraint imposed on the gossip-matrix, say W, is to respect the underlying network graph. That being the case, I find optimality w.r.t. W somewhat artificial. A given network graph can accommodate any choice of valid W. - As far as I'm aware, the applicability of such algorithms for real-life decentralized applications is yet-to-be-proven on scale.

Correctness: I haven't examined the theoretical convergence analysis in detail, but the claims seem sound. The empirical comparison seems fair and matches in breadth the venue.

Clarity: The paper is well-written and relatively easy to follow. My only suggestions: -- The rate table provided in the appendix (a standard by now, for such papers) is very convenient and is better placed in the main body to provide a quick context for the paper. -- Personally, I think some parts of the introductory sections are very repetitive and can removed and used instead to promote more interesting technical details of the analysis.

Relation to Prior Work: I find the the related work section sufficiently inclusive and useful.

Reproducibility: Yes

Additional Feedback: -- in the abstract, "The our" -- Section 4.1, perhaps further elaboration on (6) would make this section clearer. -- L231, redundant parenthesis

[Author Response · NeurIPS 2020]

We thank Reviewers (R) 1, 2 and 3 (who gave us marks 6, 7, and 8, respectively) for their pertinent remarks.

**R1+R2+R3. More details on monotone operators.** We will provide more details on the monotone operator frame-
work (for instance more elaboration around Eq. 6) allowing us to prove the theorems (we shall use the 9th page of
the camera ready if our paper is accepted). We already provided some intuitions in the appendix, e.g. in Appendix
C. Moreover, we want to recall that the definition of the space E is provided in Appendix C. We will move it to the
beginning of Section 4.1 for better understanding.

**R1+R3. Straightforward combination of existing techniques?** As R3 says, our work closes the theoretical gap
by providing the first (and so far the only) optimal first order algorithm for smooth strongly convex decentralized
optimization. We obtained this algorithm by "combining existing approaches", but the combination was far from
straightforward.

• (Scaman et al.'17) obtained MSDA by simply applying Nesterov acceleration to the dual problem. We instead
build upon recent results on the minimization of strongly convex functions under linear constraints (by a first order
algorithm, without projecting on the constraints space), see (Salim et. al.'20). Surprisingly, there are only a few
algorithms that can solve such problems at a linear rate, and they were proposed only recently.

• APAPC is obtained by applying Nesterov acceleration to the generalized forward-backward algorithm (5) for a sum
of operators $A$ and $B$ (see page 5, line 208). Although we managed to do this, this was not an easy task (to say the
least), because Nesterov acceleration does not apply to general monotone operators. Even if it it did, a naive approach
would lead to a sublinear rate $\mathcal{O}\left(\frac{1}{k^2}\right)$ because $A + B$ is not strongly monotone. On the contrary, we obtained an
accelerated linear rate (complexity $\mathcal{O}\left(\left(\sqrt{\chi\kappa} + \chi\right)\log\frac{1}{\epsilon}\right)$) in Theorem 2, which requires careful and deep theoretical
analysis of APAPC. Finally, we had to carefully design our generalized Forward Backward algorithm by choosing the
space E (see Appendix C), its inner product, and the matrix $P$ as functions of the gossip matrix $W$.

• In Appendix F we provided an algorithm provably optimal in "# of communication rounds", without using Chebyshev
acceleration. The development and analysis of this method required substantial innovation, as we explain in the paper.

Finally, we believe that the *apparent* simplicity of our approach is due to us spending a lot of time making sure the
explanations are as intuitive as possible. Many of these intuitions only became clear to us after we have done the
analysis; and we provide them for the benefit of the reader. Hence, we view the simplicity as a strength!

**R1. Experiments.** The networks chosen for evaluating the decentralized method are the ones that were used in
(Scaman et al.'17): $10 \times 10$ grid and Erdös-Rényi random network with parameter $p = 0.06$. We have now added this
detail to the paper. **Regarding the wall-clock-time comparison:** The design of our experiments was very similar to
those in (Scaman et al.'17), who assumed that local gradient computation takes one unit of time and communication
with neighbors takes time $\tau$. It's easy to observe that in this case [wall clock time] $=$ [# of gradient calls] $+ \tau \times$
[# of communication rounds]. Scaman et al. (2017) used 2 regimes in their experiments: $\tau \gg 1$ (high communication
time) and $\tau \ll 1$ (low communication time), which more or less correspond to our plots "# of communication rounds"
and "# of gradient calls". This controlled setup is sufficient to verify numerically that our theory (which expresses
the optimality of OPAPC in terms of "# of communication rounds" and "# of gradient calls") has predictive power for
experiments. Note that our "# of communication rounds" and "# of gradient calls" plots give understanding of how
the algorithms will behave with any possible $\tau$ (even if we do not know it in practice). We will also produce plots
providing "wall clock time", but these will be implementation dependent. Our focus here was not to produce highly
performing and fine-tuned software to be benchmarked in this way.

**R3. Minor comments.** We will put Table 1 in the main paper using the 9th page if the paper is accepted.

**R3. Open remark on optimality w.r.t. $W$.** In the line of works on optimal distributed algorithms by Scaman,
Hendrikx, Xiao, Bubeck, Bach and Massoulié, lower bounds are obtained by proving the existence of a "bad" gossip
matrix and "bad" functions that cannot be optimized faster than the lower bounds by any decentralized algorithm. This
includes the decentralized algorithms using the gossip matrix $W$. Therefore, if one pick a family of functions and a
gossip matrix $W$, they could be "bad" in the above sense, and one cannot beat the lower bounds by a decentralized
algorithm using $W$. However, we agree that, perhaps, the lower bounds theory for these distributed algorithms could
be a bit improved by providing lower bounds involving intrinsic properties of the graph, as in the centralized case
(there are indeed many gossip matrices for one graph).

**R3. Open remark w.r.t. real-life application.** Since OPAPC is practical and optimal, the use of OPAPC at scale is
definitely the next step in the study of OPAPC. Obviously, this goes beyond the scope of this paper, which contains
algorithm development, analysis and testing, but does not and was not supposed to have a software/system development
element. But obviously we are also very curious about its performance at scale, and plan to work on this in the future.
We are optimistic and confident that OPAPC will outperform existing approaches on average, as indicated by our
experiments.

[Meta-Review · NeurIPS 2020]

The paper makes a solid theoretical contribution to distributed optimization, and this was acknowledged in all three reviews received for the paper. Please take into account the concrete suggestions of R1 and R3 when preparing the final version.